# Nanofluids Minimal Quantity Lubrication Machining: From Mechanisms to Application

**Anxue Chu [1], Changhe Li [1,*], Zongming Zhou [2,*], Bo Liu [3], Yanbin Zhang [1], Min Yang [1], Teng Gao [1], Mingzheng Liu [1], Naiqing Zhang [4], Yusuf Suleiman Dambatta [1,5] and Shubham Sharma [1,6]**

[1]  School of Mechanical and Automotive Engineering, Qingdao University of Technology, Qingdao 266520, China; anxuechu@163.com (A.C.); zhangyanbin1_qdlg@163.com (Y.Z.); yummy0lige@163.com (M.Y.); qdlg_gt@163.com (T.G.); lmzzz654321@163.com (M.L.); ysidambatta@yahoo.com (Y.S.D.); shubham543sharma@gmail.com (S.S.)

[2]  Hanergy (Qingdao) Lubrication Technology Co., Ltd., Qingdao 266200, China

[3]  Sichuan New Aviation TA Technology Co., Ltd., Shifang 618400, China; mryhyf@126.com

[4]  Shanghai Jinzhao Energy Saving Technology Co., Ltd., Shanghai 200436, China; 13918301138@163.com

[5]  Mechanical Engineering Department, Ahmadu Bello University, Zaria 810106, Nigeria

[6]  Department of Mechanical Engineering, IK Gujral Punjab Technical University, Jalandhar 144603, India

*  Correspondence: sy_lichanghe@163.com (C.L.); 15610520185@163.com (Z.Z.)

**Abstract:** Minimizing the negative effects of the manufacturing process on the environment, employees, and costs while maintaining machining accuracy has long been a pursuit of the manufacturing industry. Currently, the nanofluid minimum quantity lubrication (NMQL) used in cutting and grinding has been studied as a useful technique for enhancing machinability and empowering sustainability. Previous reviews have concluded the beneficial effects of NMQL on the machining process and the factors affecting them, including nanofluid volume fraction and nanoparticle species. Nevertheless, the summary of the machining mechanism and performance evaluation of NMQL in processing different materials is deficient, which limits preparation of process specifications and popularity in factories. To fill this gap, this paper concentrates on the comprehensive assessment of processability based on tribological, thermal, and machined surface quality aspects for nanofluids. The present work attempts to reveal the mechanism of nanofluids in processing different materials from the viewpoint of nanofluids' physicochemical properties and atomization performance. Firstly, the present study contrasts the distinctions in structure and functional mechanisms between different types of base fluids and nanoparticle molecules, providing a comprehensive and quantitative comparative assessment for the preparation of nanofluids. Secondly, this paper reviews the factors and theoretical models that affect the stability and various thermophysical properties of nanofluids, revealing that nanoparticles endow nanofluids with unique lubrication and heat transfer mechanisms. Finally, the mapping relationship between the parameters of nanofluids and material cutting performance has been analyzed, providing theoretical guidance and technical support for the industrial application and scientific research of nanofluids.

**Keywords:** nano-cutting fluids; nanoparticles; NMQL; material

## 1. Introduction

In the metal cutting process, the cutting fluids serve multiple functions, including forming a lubricating oil film, dissipating heat, removing chips, enhancing the cutting environment, and ultimately improving product quality [1]. Nonetheless, the disadvantage of mineral oil-based cutting fluids has garnered increasing attention [2,3]. Dragičević's survey found that the cost of mineral oil-based cutting fluids can comprise 7% to 17% of the total expenses [4]. Nevertheless, the high-speed rotation of the tool will drive the surrounding air to form an air barrier layer, and lubricants have difficulty breaking through the air barrier due to the low flow rates [5]. If an insufficient amount of refined mineral

oil is used as the cutting fluid, the polycyclic aromatic hydrocarbons contained in it may cause cancer. Additionally, using mineral oil is difficult in terms of biodegradability and does not comply with ISO 14064 and the GHG Protocol [6]. Furthermore, with the rapid development of aerospace, medical equipment, etc., the requirements for processing the accuracy of key components continue to increase [7,8]. Conventional cutting techniques face challenges in attaining the requisite machining precision for certain hard-to-process materials. Exploring new cutting technologies and lubricating mediums is an important step in the development of the manufacturing industry [9–11].

In the 1990s, dry cutting technology eliminated the expenses associated with cutting fluid [12]. This technology has undergone extensive research and application in both academia and industry [5], which has introduced novel application prospects in the realm of sustainable development. This is great progress towards realizing cleaner production technology [13]. However, due to the absence of effective cooling means, it is difficult to effectively deal with the high temperature brought by the moment of material removal. Masoudi et al. [14] conducted a comparison between dry machining and wet machining in the turning of AISI 1045 steel. The study indicates that dry machining results in a higher cutting force, thermal gradient, and residual stress than wet machining. Excessive cutting forces, thermal gradients and residual stresses can adversely impact workpiece surface finish, diminish surface integrity, and potentially lead to a shortened tool lifespan [15]. In most cases, using dry machining as the main machining technology for cutting is difficult. The air-cooling process is an environmentally friendly processing technology. Liu et al. [16] conducted turning experiments on Ti-6Al-4V under dry machining and the air-cooling process. The cutting performance in the air-cooling process exhibited improvements over dry machining, yet the air-cooled cutting process introduced new issues, such as wrinkles and chipping [17]. Furthermore, the outcome of processing difficult-to-machine materials under the air-cooling process is unsatisfactory results [18]. Air-cooling processing is not the optimal approach for enhancing processing conditions [19]. The self-lubricating tool achieves the function of total automatic lubrication during the cutting process by embedding or coating materials inside its surface [20]. The ability of the tool to maintain stable cutting in harsh environments also helps to improve the sustainability and environmental friendliness of the machining process by reducing the use and disposal of lubricants [21].

MQL is a sustainable cutting lubrication method and holds significant potential for cutting applications [22]. Compressed air and cutting fluid are blended to create a fine spray of micron-scale oil droplets [23]. The high-speed air pump propels this mixture into the -tool–workpiece interface via a nozzle for cooling and lubrication [24]. This technology overcomes the hindrance posed by the "air barrier" resulting from high-speed tool rotation, ensuring efficient utilization of the cutting fluid and thorough lubrication of the -tool–workpiece interface [25]. The cutting fluid consumed by MQL is less than 10% of the flood cutting supply [26], which can achieve the same or an even better lubrication effect than traditional flood cutting. The MQL device features a straightforward design and can be conveniently positioned in proximity to the cutting machine tool [27]. However, when the amount of lubricant is reduced, some cutting fluids with poor heat transfer performance cannot meet the cooling requirements in the cutting process. Lubricants with low heat transfer efficiency, combined with compressed air, can only extract a restricted quantity of heat within the machining zone, thereby failing to attain the projected cooling efficacy [28]. Heat accumulation can deteriorate the processing conditions, burn the surface of the workpiece, and shorten the life of the tool [29]. The urgent need for cooling media with excellent thermal conductivity is gradually emerging [30]. Abdullah et al. [31] evaluated the feasibility of the MQL solution for Strenx 900 steel processing. Mallick et al. [32] examined the improvement effect of MQL on turning performance. Sun et al. [33] compared the milling process under dry and MQL lubrication methods; under the MQL lubrication method, the milling force and milling temperature decreased. Figure 1 is a brief review of the development of cutting lubrication technology.

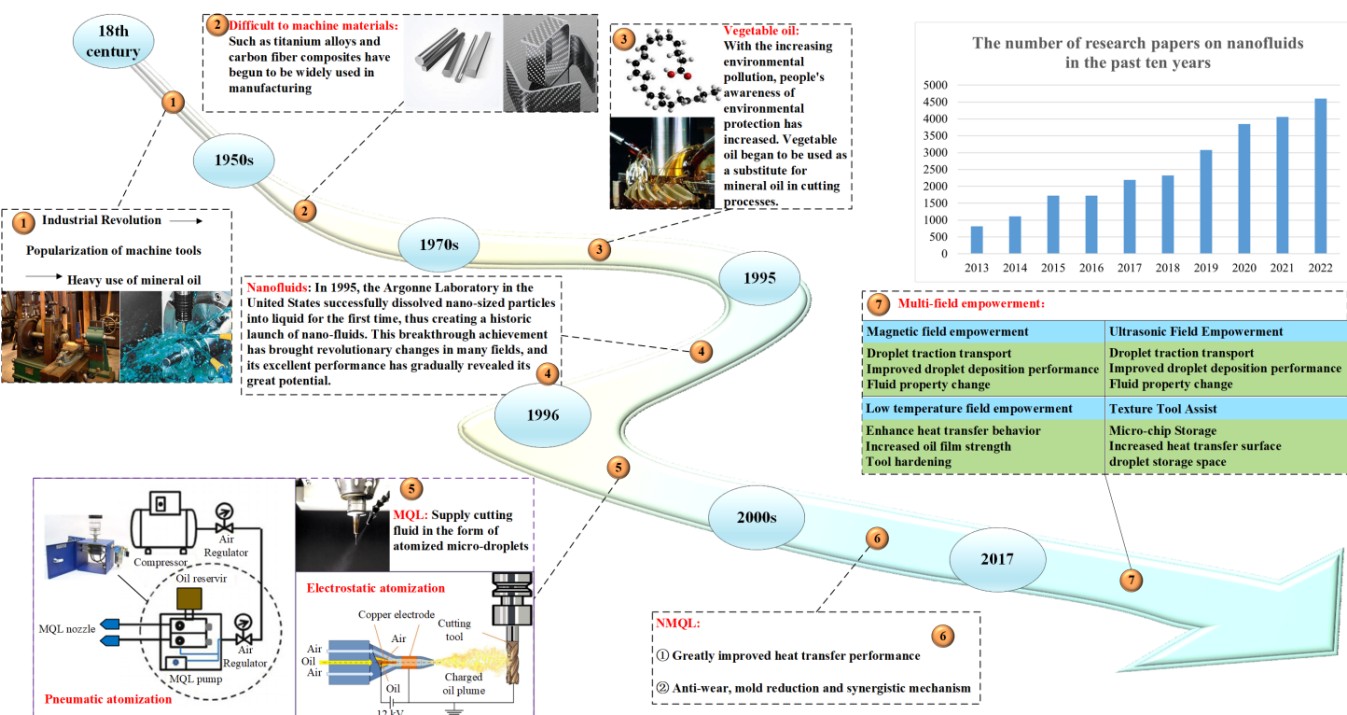

**Figure 1.** A short chronological review of developments in cutting lubrication technology.

The concept of nanofluids was first proposed by Choi of Argonne Laboratory in the United States in 1995 [34]. Nanofluids are affected by the type, size, shape, preparation method and additives of nanoparticles. Compared with traditional heat transfer media, nanofluids have better performance in terms of thermal conductivity, viscosity, specific heat, etc. [35]. Choi added a 1% volume fraction of nanoparticles to the traditional heat transfer medium, and the thermal conductivity of the fluid increased by approximately two fold [36]. Nanofluids find extensive applications in thermal management, the automotive industry, manufacturing, aerospace, and various other domains [37–39]. Diglio et al. [40] studied the application of nanofluids as heat carriers in heat exchangers. Nanoparticles range in size from 1 to 100 nm, solving the technical challenges associated with particle deposition and microchannel plugging often encountered in micron-scale materials applications [41]. Said et al. [42] successfully applied nanofluids to flat-plate solar collectors. Compared with macroscopic particles, the application process of nanoparticles has better performance [43,44].

NMQL is an efficient and environmentally- friendly precision lubrication technology that combines MQL technology with the use of nanofluid as a cutting fluid [45]. Solids possess significantly higher thermal conductivity compared to liquids [46]. Introducing nanoparticles into cutting fluids to prepare nano-cutting fluids can overcome the limitations of MQL heat transfer technology [47,48]. Moreover, owing to the distinctive structure of nanoparticles, upon entering the -tool–workpiece interface, these particles can unleash a synergistic impact that combines anti-wear and friction reduction via the ball effect, the film-forming effect, the filling effect, and the polishing effect [49]. Sui et al. [50] used MoS$_2$-castor oil as a nano-cutting fluid in the process of grinding cemented carbide. In comparison to dry machining, the decrease in grinding temperature achieved through NMQL was double that of pure oil cutting fluid MQL. Zhou et al. [51] conducted a review on the utilization of nano-cutting fluids in the cutting of nickel-based alloys. Numerous experimental results indicate that NMQL enhances surface quality, extends tool lifespan, lowers processing temperature, and boosts processing efficiency in nickel-based alloy machining. Revuru et al. [52] deliberated on the potential of nano-cutting fluids to mitigate the challenges posed by the high hardness and low thermal conductivity of titanium alloys

in cutting processes. Lawal et al. [53] evaluated the sustainability of nano-cutting fluids in the cutting process of carbon fiber-reinforced plastics (CFRP). Nanofluids not only reduce the necessary cutting fluid volume in the machining process but also curtail the defect rate of completed CFRP workpieces, fostering the wider implementation of CFRP. While the viability of nanofluids in cutting various materials has been extensively validated [54,55], a comprehensive inquiry is still required to elucidate the correlation between different processing materials and the optimal selection of nanofluids.

The article structure is shown in Figure 2. Firstly, this review outlines the characteristics of nanofluids, including preparation, stability, and their thermophysical properties. Secondly, the unique cooling and lubricating mechanism of nanofluids in cutting processing is expounded, including the mechanism of anti-friction and anti-wear, the mechanism of enhanced heat transfer and the mechanism of atomization and penetration endowed by MQL. Additionally, an assessment is conducted on the cutting performance of various materials when subjected to NMQL processing under diverse boundary conditions. The mapping relationship between material processing and nanofluid selection is established, which provides a reference for industrial applications.

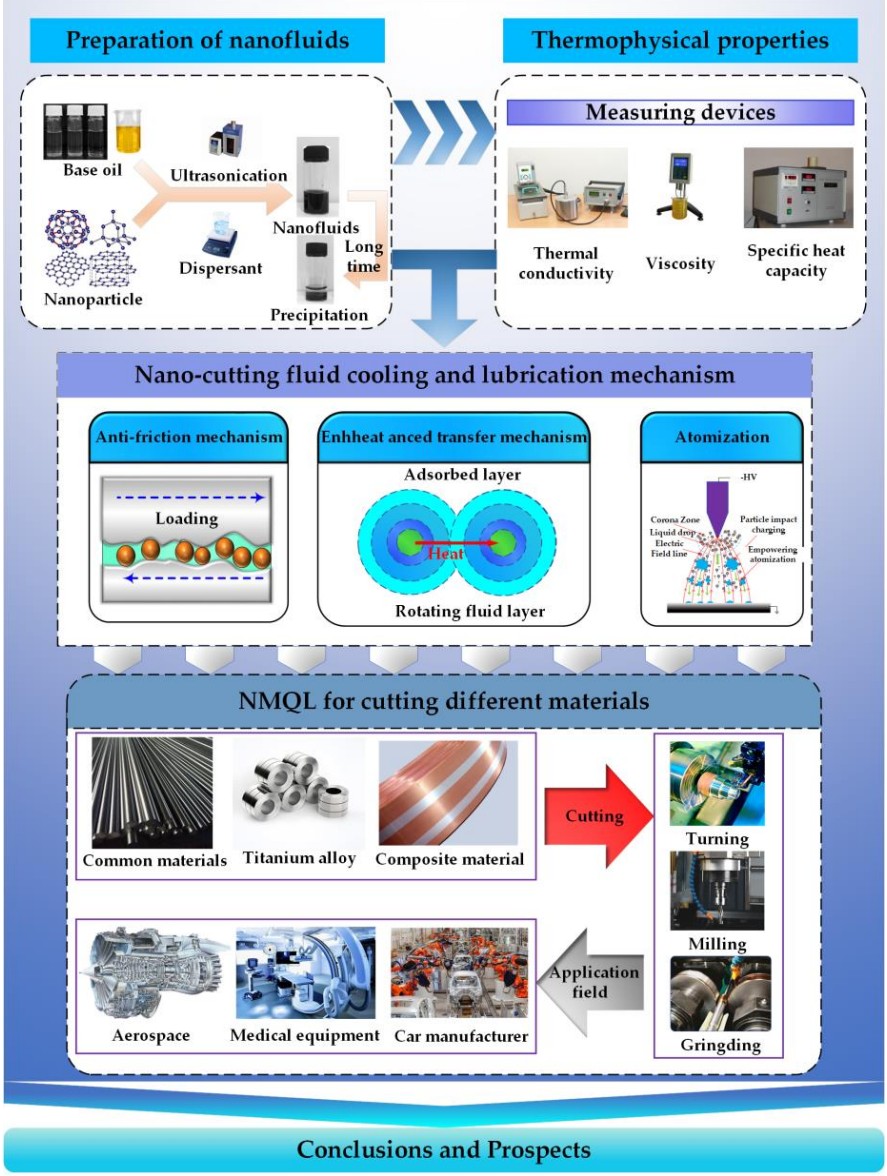

**Figure 2.** Structure of this paper.

## 2. Preparation of Nanofluids

Nanofluids, which consist of -solid–liquid mixtures, are created by evenly dispersing nanoparticles within a base fluid [56]. The remarkable performance of nanofluids heavily relies on the characteristics of the base fluid and the nanoparticles. Prerequisites for the application of nanofluids include thorough mixing during their preparation as well as long-term stability throughout the storage duration [57].

### 2.1. Base Fluid Selection

Although mineral oil lubricants have formed a complete system of use in cutting production and processing, with the development of human society and the increasing public awareness of environmental protection, the pollution caused by mineral-based oils must be fundamentally reversed [58]. In recent years, degradable, renewable vegetable oils and low-cost, easy-to-clean water-based lubricants have become popular alternatives to mineral oils [59,60]. The advantages and disadvantages of three different cutting fluids are shown in Figure 3. This section, the usage characteristics of two new environmentally friendly cutting fluids will be introduced in detail.

| Cutting fluids | Advantages | Disadvantages |
|---|---|---|
| Mineral oils-based lubricants | Good lubrication and rust prevention | Ecological and health issues |
| | Mature technology | Wastewater disposal cost issues |
| | Chemically stable | Difficult to apply to extreme friction |
| Vegetable oils based cutting fluid | Excellent lubricity | Insufficient heat transfer performance |
| | Environmentally friendly and renewable | Easily oxidized at high temperatures |
| Water-based lubricants | Excellent heat transfer performance | Strong corrosive |
| | Cheap | Poor lubrication |

**Figure 3.** Commonly used cutting fluid advantages and disadvantages.

### 2.1.1. Vegetable-Based Oil

Vegetable oils have gained wide attention due to their biodegradable and renewable properties [61]. Vegetable oil raw materials are renewable, and they absorb more carbon dioxide during plant growth than they release, helping to balance the carbon dioxide generated by industrial production [3]. Furthermore, when vegetable oil is applied as a cutting fluid in industrial production, the waste oil can be decomposed into carbon dioxide, hydrogen gas, and biomass, forming a complete carbon cycle [62,63]. In contrast, the use of mineral oil only increases carbon dioxide emissions, contributing to additional greenhouse effects [64].

Vegetable oils are composed of glycerol and fatty acids. The molecular structure of fatty acids includes carbon chains and polar groups. Substantial differences exist among various fatty acids [65]. Saturated fatty acids consist solely of single-chain carbon atoms. Conversely, fatty acids with C=C double bonds are referred to as unsaturated fatty acids. The C=C double bond has the capability to induce curvature in a carbon chain. The schematic structural diagram of saturated fatty acids and unsaturated fatty acids is shown in Figure 4a, in which the carbon chain of unsaturated fatty acids is bent, which is caused by the presence of C=C [49]. Greater unsaturation in vegetable oil correlates with higher friction coefficients. Reeves et al. [66] elucidated variations in the processing characteristics of distinct vegetable oils through the construction of a model for assessing the extent of fatty acid unsaturation. The study revealed a correlation index of 0.830 between the coefficient of friction and the level of unsaturation. Figure 4b shows the distribution proportions of saturated fatty acids, monounsaturated fatty acids and polyunsaturated fatty acids in eight common vegetable oil lubricants. Figure 4c shows the degradation temperature of the above-mentioned vegetable oil detected in a nitrogen environment. The more carbon atoms,

the stronger the molecular cohesion, the higher the viscosity, and the better the lubricating performance. Additionally, polar groups can influence the lubrication characteristics of vegetable oils. The majority of vegetable oil tails is -COOH or -COOR, but a small number of vegetable oils have -OH groups at the tail end of their carbon chains [63]. Li et al. [67] investigated and contrasted the grinding performance of seven distinct vegetable oils. Owing to its special polar group -OH, castor oil displays the highest viscosity, the least grinding force, yet the greatest grinding temperature and energy ratio coefficient; this is because high-viscosity vegetable oils provide better lubrication and a higher load-carrying capacity in the cutting zone.

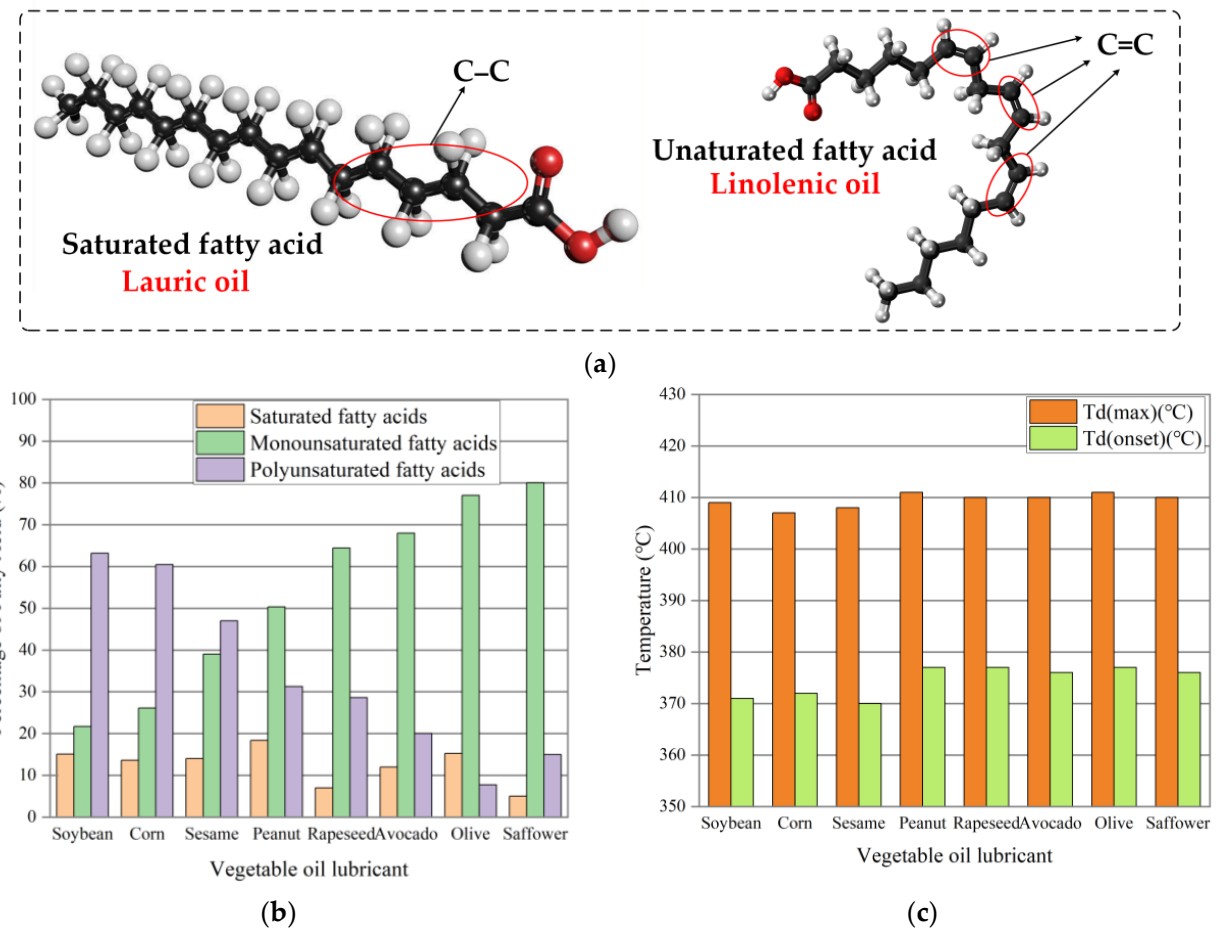

**Figure 4.** (**a**) Carbon chain structure and degree of unsaturation; (**b**) fatty acid concentration for different vegetable oils; (**c**) thermal degradation temperature of vegetable oils (in nitrogen) (reproduced from [66], with permission from Elsevier).

In terms of MQL experimental research, vegetable oil and its esterification products are the main choices of lubricant. Khan et al. [68] reported the performance of vegetable oils as lubricants in MQLT AISI 9310 turning processes. Using vegetable oil as a lubricant significantly reduces flank wear, increases tool life and improves surface finish compared to dry machining and submerged cooling. Gajrani et al. [69] developed an environmentally friendly plant-based green cutting fluid and evaluated its machining performance in turning AISI H-13 steel; this self-made vegetable oil cutting fluid significantly reduced the cutting force, feed force, friction coefficient and surface roughness. High viscosity will affect the heat transfer ability of vegetable oil, and excessive heat accumulation will deteriorate the surface quality of machined parts [70]. Enhancing the heat transfer capability of vegetable oil is a prerequisite for large-scale industrial applications. Adding nanoparticles is the main means to enhance the heat transfer performance of vegetable oil [71].

### 2.1.2. Water-Based Lubricants

Water-based lubricating fluids are also good lubricants commonly used to replace mineral oil. Water-based lubricants are easy to clean, have little environmental pollution, and do not undergo saponification due to high temperatures [72]. Water has high cooling properties. However, the viscosity of water is very low, and using water alone as a cutting fluid has poor lubricating properties. Blending additives through various processes is a common means of enhancing performance [73]. In general, additives can act as antioxidants, disinfectants, etc. Commonly used additives include foaming agents, corrosion inhibitors, cleaning agents, friction modifiers, wear modifiers, metal deactivators, and viscosity modifiers [74]. In recent years, nanoparticles have received much attention from researchers as an alternative additive. Table 1 is the practical situation of nanofluid water-based cutting fluids.

**Table 1.** The improvement effect of nanoparticles on water-based lubricants.

| Researchers | Nanoparticles | Work Condition | Effect |
|---|---|---|---|
| Gu et al. [75] | $TiO_2$ | Milling 235 steel | The wear scar diameter and friction coefficient decrease after $TiO_2$-water nano-lubricant processing. |
| Cui et al. [76] | $TiO_2$, $SiO_2$ | Tribological experiments on ceramic materials | $SiO_2$ forms a more uniform protective film in water, and the tribological performance is better. |
| Liu et al. [77] | Graphene (GR) | Steel ball—plate friction test | Compared with pure deionized water lubricant, the friction coefficient and wear rate were reduced by 54.6% and 45.0%, respectively, by adding GR. |
| Xie et al. [78] | $SiO_2$/GR | Rolled magnesium alloy sheet | $SiO_2$/GR hybrid nanofluid is more conducive to tribological properties and rolling lubrication performance, $SiO_2$ prevents assembly between GR layers. |
| Najiha et al. [79] | $TiO_2$ | Milling AISI D2 tool steel | $TiO_2$ Nano-lubricant reduces workpiece adhesion and improves tool life. |

Through the addition of nanoparticles, the shortcomings of poor anti-friction performance, high corrosion, and low viscosity can be overcome to the greatest extent. Further exploration of the use and development of additives has also created lubricants that can be used in a variety of applications to meet processing requirements in extreme environments.

### 2.2. Nanoparticles

Nanoparticles are advanced materials characterized by their nanoscale dimensions. Commonly employed synthesis methods encompass physical techniques, chemical synthesis and biological cultivation methods [80]. The specific preparation method of nanoparticles is shown in Figure 5.

### 2.2.1. Type of Nanoparticle

According to the type of raw materials, the current classification of nanoparticles includes metal nanoparticles, metal oxide nanoparticles, and non-metal nanoparticles, which are collectively referred to as single nanoparticles. Corresponding to it is hybrid nanoparticles, which are mainly two-phase mixed [81].

The production cost of metal nanoparticles is relatively high, and they are susceptible to uneven oxidation. Therefore, they are not suitable for large-scale industrial production [82]. Metal oxide nanoparticles are an economical alternative to metal nanoparticles and are often used in industrial application, so $Al_2O_3$, $TiO_2$, ZnO, and CuO [83].

Commonly used non-metallic nanoparticles include carbon nanotubes (CNTs), GR, silicide (SiC, $SiO_2$) and nano-diamond (ND). Carbon nanotubes can be divided into single-walled CNTs (SWCNTs), double-walled CNTs (DWCNTs) and multi-walled CNTs (MWCNTs) according to their structures. MWCNTs have been widely considered by researchers

because of their special structure that can effectively reduce the sliding friction in the cutting zone [84]. GR has a unique layered structure and chemical inertness, which enables it to stably exert friction and anti-friction effects at the -tool–workpiece interface [85]. $SiO_2$ and SiC have strong adsorption due to the presence of hydroxyl groups, and can form a stable physical adsorption film, thus playing the role of reducing friction and antifriction [86]. ND is a non-metallic nanoparticle with extremely high hardness and a stable structure, which is an excellent nano-additive in the process of material removal [87]. Table 2 lists relevant research on nanoparticles improving the performance of cutting fluids.

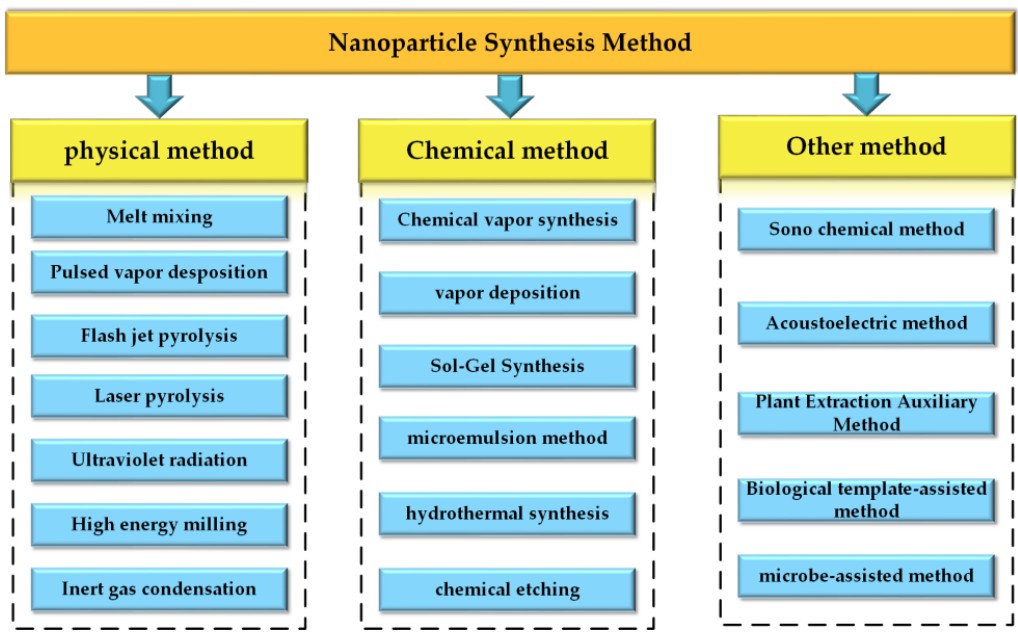

**Figure 5.** Preparation method of nanoparticles.

**Table 2.** Nanoparticles Improve the Performance of Cutting Fluid.

| Researchers | Nanoparticles/ Based Fluids | Processing Conditions | Effect |
|---|---|---|---|
| Agunsoye et al. [86] | $SiO_2$/RHA | Turning Aluminum Alloy | Nano-cutting fluid contributed 92.96%, 88.59%, and 93.40% in reducing cutting force, flank wear, and improving surface finish. |
| Tong et al. [88] | $Al_2O_3$/Palm oil | Milling | The $Al_2O_3$-palm oil nanofluid exhibited the best heat transfer and lubrication properties at a concentration of 1.0 vol% and a particle size of 20 nm. |
| Kumar et al. [89] | $Al_2O_3$, $TiO_2$/ Deionized water (DW) | Turning AISI D2 | The flank wear of the $TiO_2$ nanofluid tool is 29% lower than that of $Al_2O_3$, the cutting temperature is 9.7% lower, and *Ra* is 14.3% lower. |
| Hamid et al. [90] | $TiO_2$/Soluble oil | Rotary drilling | $TiO_2$-soluble oil nanofluid decreased the operational temperature of drilling procedure. |
| Edelbi et al. [91] | $Al_2O_3$, ZnO/LRT-30 oil | Milling of Ti-3Al-2.5V | Compared with $Al_2O_3$ nanofluids, the cutting temperature and surface roughness of ZnO nanofluid are 1% and 2% lower, respectively. |
| Huang et al. [92] | MWCNTs/Cutting oil | Milling AISI P21 and 1050 | MWCNTs significantly reduce the cutting temperature. 9% improvement in surface roughness. |
| Zhang et al. [93] | SiC/Rapeseed oil | Turning 40Cr | NMQL reduces wear by 55.1%, cutting temperature by 41.5% and surface roughness by 19.2%. |

The hybrid nanoparticles enhance heat transfer performance better than the single nanoparticles. Chen et al. [94] used MWCNTs to modify the surface of Ag nanoparticles. Compared with MWCNT nanofluids, the hybrid nanofluids exhibit higher thermal conductivity. Bioucas et al. [95] attributed this phenomenon to the synergistic effect of mixed nanoparticles. In actual cutting applications, Barewar et al. [96] carried out machining with superalloy Inconel 718 using Ag/ZnO-ethylene glycol (EG) as a nano-cutting fluid. The hybrid nanofluid improved heat dissipation, resulting in an 8.56% reduction in the cutting temperature compared to pure ethylene glycol. Sharma et al. [97] employed an $Al_2O_3$ and $MoS_2$ hybrid nanofluid as a cutting fluid during the turning of AISI304 stainless steel; this hybrid nano-cutting fluids excels at reducing cutting forces and improving surface integrity.

### 2.2.2. Shape of Nanoparticles

According to the geometric particle structure and size of nanoparticles, nanoparticles can be divided into spherical (zero-dimensional), threadiness (one-dimensional), and stratiform (two-dimensional) nanoparticles. Different structures show different shape effects [83]. Figure 6 shows the difference between the structure and mode of action of the three shapes of nanoparticles.

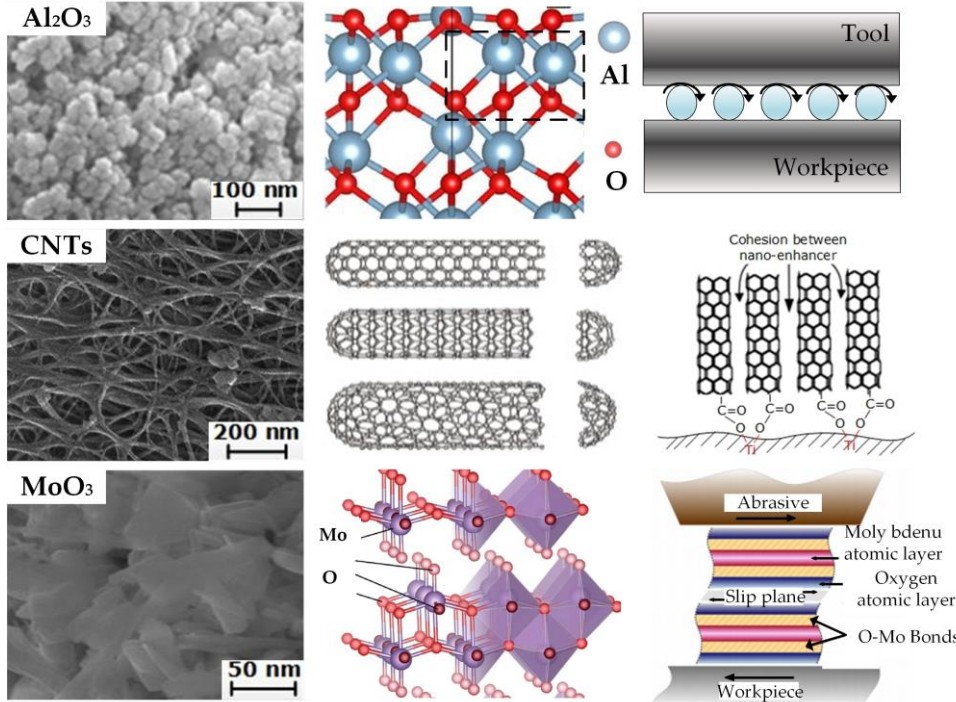

**Figure 6.** Schematic diagram of the structure and mechanism of action of nanoparticles with three shapes.

$Al_2O_3$ is a typical spherical nanoparticle. This type of nanoparticle can perfectly adapt to the "rolling effect" and act as a "bearing" structure for the -tool–workpiece interface, reducing interface friction and cutting force to improve surface integrity as well as reducing the tool lifespan [98]. In contrast, spherical nanoparticles often exhibit relatively low thermal conductivity, and improving the thermal conductivity of spherical nanoparticles is the key goal of breakthroughs in the current research field [99].

The aspect ratio of threadiness nanoparticles has a great relationship with their thermal conductivity enhancing ability. Li et al. [100] explored the relationship between the shape effect of non-spherical nanoparticles and thermal conductivity and convective heat transfer, finding that for nanorods and ellipsoids with small aspect ratios, the thermal conductivity enhancement ability is obvious. On the contrary, for nanoparticles with a large aspect ratio, only carbon nanotubes exhibit good thermal conductivity.

Due to the low shear strength between adjacent layers, stratiform nanoparticles can act as nanoscale sliders at the -tool–workpiece interface [48], which can reduce the friction between the two interfaces and reduce the cutting force and cutting temperature during the cutting process, thereby improving the surface quality of the workpiece.

Spherical nanoparticles are excellent in terms of lubricating performance, but insufficient in terms of cooling performance. Threadiness nanoparticles have excellent heat transfer performance, and the performance enhancement of a sort of nanoparticle often focuses on a certain aspect. Hybrid nanoparticles may compensate for this problem, relying on the synergistic effect between different particles to show better performance in improving cooling and lubrication performance. Sharma et al. [101] aimed to address the insufficient heat transfer performance of spherical $Al_2O_3$ nanofluids, prepare $Al_2O_3$-MWCNT nanofluids and conduct comparative experiments. The results showed that the addition of MWCNTs improved the thermal conductivity of nanofluids. Shape distribution of common nanoparticles shown in Table 3.

**Table 3.** Common nanoparticles and their classification and shape.

| Type | Nanofluids | Shape |
|---|---|---|
| Metal oxide nanoparticles | $Al_2O_3$ | spherical |
| | $SiO_2$ | spherical |
| | $ZrO_2$ | spherical |
| | $CuO_2$ | threadiness |
| | $MnO_2$ | threadiness |
| | $TiO_2$ | threadiness |
| | $ZnO_2$ | threadiness |
| | $MoO_3$ | stratiform |
| Non-metallic nanoparticles | SiC | spherical |
| | ND | spherical |
| | CNTs | threadiness |
| | GR | stratiform |
| | HBN | stratiform |
| | $MoS_2$ | stratiform |

### 2.3. The Preparation Method

The stable preparation of nanofluids is a crucial prerequisite for both experimental research and industrial applications. Researchers aim to develop techniques that can create nanofluids capable of achieving long-term storage while retaining their excellent properties [102]. Currently, two common methods are employed for nanofluid preparation: the one-step method and the two-step method. The advantages and common problems of these two commonly used methods are listed Table 4.

### 2.3.1. One Step

The one-step method is a nanofluid preparation method in which the synthesis and dispersion of nanoparticles are completed simultaneously. Compared with the two-step method, the one-step method eliminates several intermediate steps and simplifies the preparation process; the specific preparation process is shown in Figure 7. This method offers better stability compared to the two-step process. However, it is only suitable for small-scale applications and base liquids with low evaporation pressure. Zhu et al. [103] successfully prepared stable nanofluids using copper nanoparticles and ethylene glycol as the base liquid through the one-step process. They evaluated the stability of the nanofluids using infrared analysis and transmission electron microscopy. Parsa et al. [104] synthesized

silver nanoparticles in water, achieving volume fractions ranging from 1% to 5% and demonstrating stability for up to one month. Ratnasari et al. [105] achieved good stability for one month using a one-step process to synthesize nanofluids in polyethylene glycol. Although the one-step process offers better stability compared to the two-step process, it is limited to extracting a limited number of nanoparticles and preparing a limited volume of nanofluids, making it unsuitable for large-scale production.

**Table 4.** Advantages and disadvantages of the one-step and two-step methods.

| Process Method | Advantage | Shortcoming |
| --- | --- | --- |
| One-step method | Less agglomeration, high stability | —— |
| | No storage, drying steps required | Excessive deposition of residual reactants |
| | No oxidize | For low-vapor-pressure base fluids only |
| | No redistributable requirements | Cannot be mass-produced |
| | No transport required | —— |
| Two-step method | Easy and cheap to make | Rapid condensation/Rapid settling |
| | —— | Requires surfactant or functionalization treatment |
| | Suitable for oxide nanoparticles | High surface energy, increased self-weight |

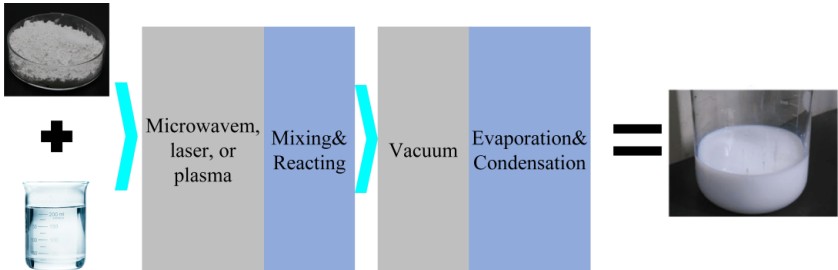

**Figure 7.** One-step production specification.

### 2.3.2. Two Steps

The two-step process involves first preparing nanoparticle powders using physical, chemical, or biological methods, and then dispersing the nanoparticles into the base liquid using techniques such as ultrasonic magnetic stirring, ultrasonic mixing, or high-shear processing. The concept and process of the two-step method and the related equipment involved are shown in Figure 8. This method is simple and cost-effective, making it suitable for large-scale industrial production [106]. However, in the powdered state, the Van der Waals force between particles is much greater than the electrostatic repulsion force, making it challenging to fully disperse agglomerates in the base liquid, and the thermal conductivity of the base liquid may not be significantly enhanced [107].

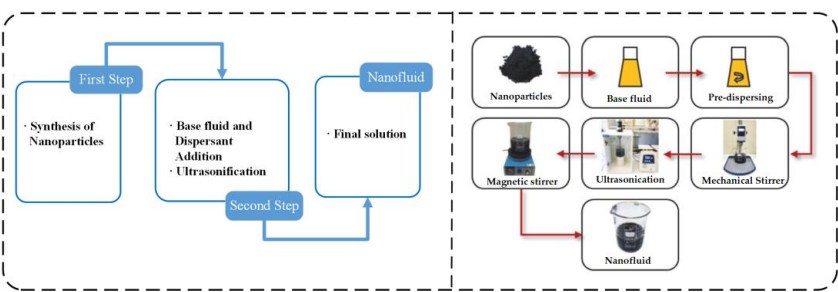

**Figure 8.** Two-step production specification (reproduced from [108], with permission from Elsevier).

To prepare stable nanoparticles, researchers not only extended the stirring and mixing time but also added surfactants to the carrier fluid to increase the stability of the nanofluids. Judran et al. [109] synthesized MgO nanoparticles in deionized water with volume concentrations ranging from 0.05% to 0.25%. The mixture of nanoparticles and base liquid was mechanically stirred for 30 min and then subjected to ultrasonic treatment, with durations ranging from 45 min to 180 min. All samples exhibited stability for more than 15 days, with better stability observed with increasing ultrasonic treatment time. The nanofluid with a volume fraction of 0.15% and an ultrasonic treatment duration of 180 min achieved a 25.08% increase in thermal conductivity. Sharma et al. [110] used an improved two-step method to prepare long-term stable CNT-water nanofluids. SDBS was used as a surfactant to enhance stability. The mixture of base liquid, surfactant, and nanoparticle was stirred using a magnetic stirrer for 10 min and then subjected to 1 h of ultrasonic treatment. The resulting nanofluids exhibited stability for over 15 months.

The two-step method is the most cost-effective way to prepare nanofluids and is therefore suitable for large-scale production of nanofluids. Using this method, nanofluids of different materials and sizes can be directly prepared, and better particle size control can be obtained, so that the prepared nanofluid has a more uniform particle distribution. However, the two-step method is more complicated and requires more time and experimental equipment. In short, both the single-step method and the two-step method have their own advantages and disadvantages. In specific applications, it is necessary to select an appropriate preparation method according to different experimental requirements and usage conditions of nanofluids. It is recommended that a two-step method be used to prepare the required nano-cutting fluid in the field of mechanical processing.

### 2.4. Stability of Nanofluids

The higher surface energy and surface area of nanofluid determine its excellent heat transfer performance, but it also makes dispersed nanoparticles very easy to agglomerate [111]. During the cutting process, the agglomeration of nanoparticles will not only affect the processing performance of the cutting fluid, but may even cause irreversible scars to the workpiece. Maintaining long-term stability of nanofluids is a prerequisite for cutting applications.

#### 2.4.1. Stability Mechanism

The stability of nanofluids can be explained according to the DLVO theory [112], which believes that there are Van der Waals attractive force and electrostatic repulsive force between the particles, as shown in Figure 9a. These two forces can produce two diametrically opposite motion trends of aggregation and separation between particles. If the Van der Waals attractive force is bigger than the electrostatic repulsive force, the particles will agglomerate and settle in the base fluid. On the contrary, if the electrostatic repulsive force is bigger than the Van der Waals attractive force and electrostatic repulsion, the nanofluid can remain stable [37]. Therefore, to obtain stable nanofluids, the repulsive force between particles must dominate. The layer on the surface of the particle can be divided into two parts: a stern layer with tightly bound negative charges and a diffuse layer with loosely scattered positive charges on the outer surface [83]. They are called electric double layers, and are shown in Figure 9a. The potential difference between the two layers is represented by the zeta potential. There is a positive correlation between the zeta potential value and stability. The higher the zeta potential value, the more stable the performance of the nanofluid, but this is not the only determining factor. It should be noted that if the particle and the base fluid have a significant density difference, even with a high zeta potential value, precipitation will occur. Steric stability and electrostatic stability jointly determine the stability of nanofluids. Figure 9b shows the steric stability and electrostatic stability of nanoparticles.

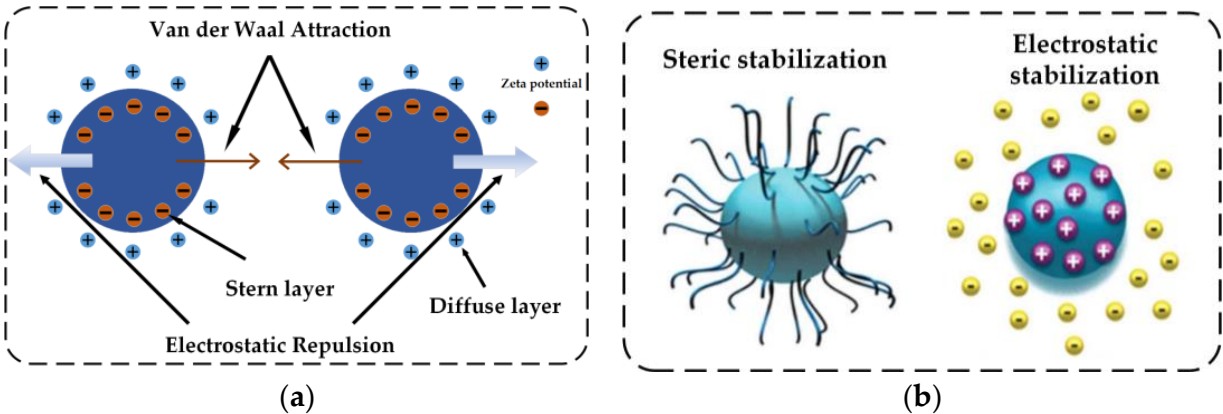

**Figure 9.** (**a**) Van der Waals force and electrostatic repulsion (reproduced from [113], with permission from Elsevier); (**b**) electrostatic stability and steric stability.

The size and shape of the nanoparticle itself are important factors affecting the stability of the nanofluid, and the size of the particle determines the size of the Van der Waals force and the electrical repulsive force [114]. The smaller the particle size, the greater the surface energy of the particles, which increases the possibility of nanoparticle aggregation. The different shapes of nanoparticles will lead to different action points and directions of the Van der Waals force and electrostatic repulsion force of the particles [115]. He et al. [116] studied the effect of particle size on the stability of hematite nanoparticles, and the results proved that as the particle size decreases, the aggregation tendency of particles becomes stronger. For the same pH value and nanoparticle concentration, the smaller the particle size, the higher the degree of aggregation. Kim et al. [40] observed that particle shape had a strong influence on the stability of the water-based bohemite aluminum nanofluids, and explained this effect to be due to the strong dependence of electrostatic repulsion on particle shape, but Van der Waals forces are hardly affected.

The pH value of the nanofluid is closely related to the zeta potential value, and the pH value will affect the surface charge density of the nanoparticles, thereby affecting the stability of the nanofluid [85]. At the isoelectric point, individual particles have no net charge, and the zeta potential value is close to zero. Nanofluidic suspensions close to their isoelectric point are considered unstable. The pH also affects the size and shape of the nanoparticles during their preparation, thereby affecting the stability of the nanofluid. Li et al. [117] investigated the effect of pH on the size and shape of the synthesized Ag nanoparticles. In the alkaline solution with a pH value of 11, the morphology of the Ag nanoparticles is basically spherical, and the size distribution is uniform, while in the neutral medium, the shape of the synthesized Ag nanoparticles varies from spherical to rod-like, and the size distribution is different.

An increase in the concentration of nanoparticles leads to a smaller interparticle spacing, which destroys the interparticle steric stability. Chakraborty et al. [118] observed that at a 40 ppm nanoparticle concentration, Cu-Zn-Al layered double hydroxide nanofluid remained stable, while at 240 ppm the nanofluid suspension showed signs of precipitation. The increase in particle concentration increases the size of particle clusters, the particle distance is smaller, and the Van der Waals gravity is enhanced, which directly affects the sedimentation velocity. Hong et al. [119] reported that the average cluster size of Fe-ethylene glycol nanofluid increased from 1.2 μm to 2.3 μm with time. Nanofluids have a higher tendency to aggregate as the concentration of nanoparticles increases.

2.4.2. Stability Enhancement Technology

At present, there are two main dispersion methods to improve the stability of nanofluids, physical dispersion and chemical dispersion. Such as in Figure 10, physical dispersion includes magnetic stirring, ultrasonic dispersion [120] and chemical methods include

adding surfactants, adjusting PH value [121]. A single physical dispersion or chemical dispersion cannot fully and uniformly disperse nanoparticles in the solution. Two or more methods must be used simultaneously in the preparation process to ensure the stability of the prepared nanofluid.

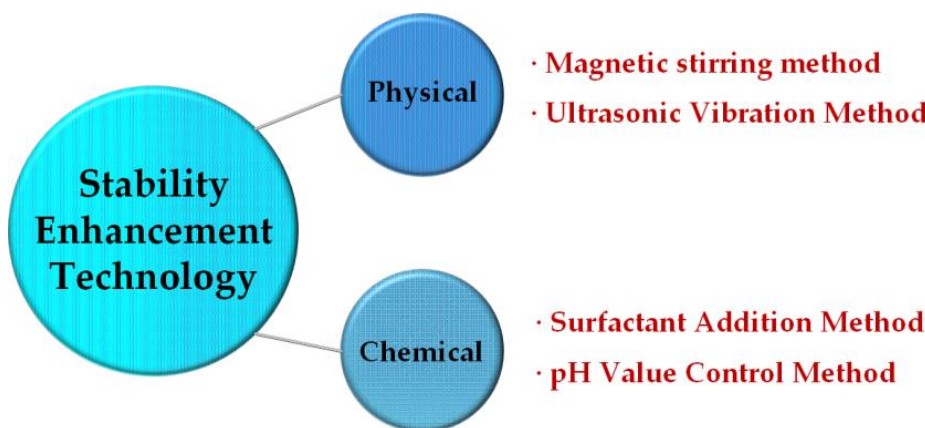

**Figure 10.** Stability enhancement technology.

The principle of magnetic stirring is to use a rotating magnetic field to reduce particle clusters to ensure the stability of nanofluids. Ultrasonic dispersion technology refers to the generation of supersonic waves through ultrasonic oscillations to disperse nanoclusters into small particles and evenly disperse them into the base liquid. Zheng et al. [122] used a two-step method to prepare GR nanofluids under different ultrasonic powers, and evaluated the effects of ultrasonic power and time on the dispersion stability of the fluid by using the relative concentration (RC) as a characterization method. Figure 11 shows the changes in RC of 0.01 wt.% and 0.1 wt.% nanofluids when they are in the stationary state. It can be seen that RC is more sensitive to sonication time than power.

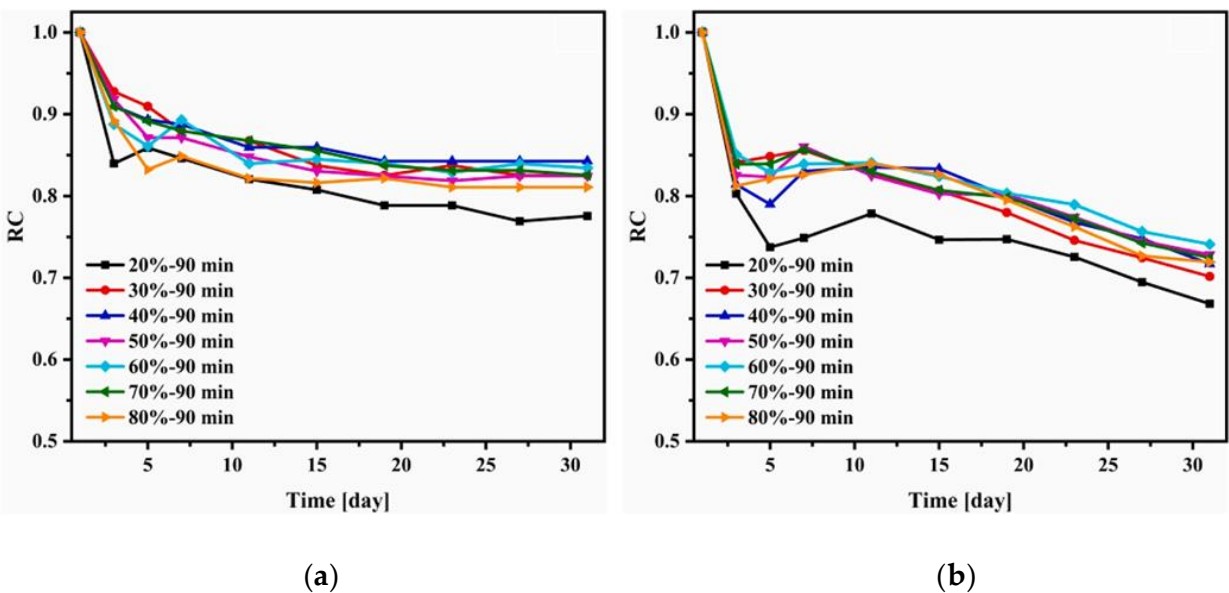

(**a**)                                                                  (**b**)

**Figure 11.** RC of nanofluids prepared with different ultrasonic power changes with time: (**a**) 0.01 wt.%; (**b**) 0.1 wt.%. (reproduced from [122], with permission from Elsevier).

Adding surfactants is a common method to improve the stability of nanofluids. Surfactants can be adsorbed between the -solid–liquid surface of nanoparticles. Interfacial steric hindrance is formed so that the particles will not cluster and settle due to mutual attrac-

tion [65]. Surfactants are mainly divided into four types: an-ionic surfactant, cationic surfactant, no-ionic surfactant and amphoteric surfactants [123]. The classification of common surfactants is shown in Table 5. The use of surfactants not only increases the stability time of nanofluids, but also improves the thermal conductivity of nanofluids. Seyhan et al. [124] used gum arabic as a surfactant, Ag as nanoparticles, and water, ethylene glycol, and n-ethane as base fluids to prepare nanofluids. The result obtained is that the gum arabic surfactant does not affect the thermal conductivity of ethylene glycol and ethane, but reduces the thermal conductivity of water-based nanofluids. Tiwari et al. [125] used six different surfactants (CTAB, DDC, SDBS, SDS, GA, and PVP) to prepare $CeO_2$+MWCNT/water-based nanofluids. It is not that the more surfactants used, the more stable the nanofluids are. The optimal addition ratio of surfactants to hybrid nanofluid is 3:2. A ratio higher than this concentration will increase the cost and decrease the stability, and will not further increase the thermal conductivity. Experimental observation shows that CTAB has the best stabilization effect within one month, and SDBS has the best stabilization effect when the time is extended to three months.

**Table 5.** Commonly used surfactants [126].

| Classification | Surfactant | |
| --- | --- | --- |
| An ionic surfactant | Sodium dodecyl sulfate | SDS |
| | Sodium dodecylbenzene sulphonate | SDBS |
| | Ammonium lauryl sulfate | ALS |
| | Potassium lauryl sulfate | PLS |
| Cationic surfactant | Cetrimonium chloride | CTAC |
| | Cetyl trimethyl ammonium bromide | CTAB |
| Non-ionic surfactant | Tween-20 | |
| | Tween-80 | |
| | Span-80 | |
| | Polyvinyl pyrrolidone | PVP |
| | Triton X-100 | |
| | Oleyl amine | |
| Amphoteric surfactant | Lecithin | |
| | Hydroxysultaine | |

## 3. Thermophysical Properties of Nanofluids

Compared with the basic fluid, the addition of nanoparticles makes the nanofluid exhibit completely different thermophysical properties. The thermal conductivity, viscosity, specific heat capacity and surface tension of nanofluids greatly affect the cutting performance [127]. This section will introduce the performance and practical model research of different nanofluids in related properties.

### 3.1. Thermal Conductivity

Thermal conductivity is the most important thermophysical property of nanofluids. Thermal conductivity is defined as the rate at which heat is transferred by conduction. The addition of nanoparticles can improve the thermal conductivity of the base fluid. The Brownian motion of the nanoparticles and the micro-convection between the nanoparticles and the base fluid accelerate the transfer of heat between the nanoparticles and improve the thermal conductivity of the nanofluids [128]. The following will review the thermal conductivity in terms of the influencing factors and model.

### 3.1.1. Influencing Factors

*Base Fluid:* The thermal conductivity of nanofluids is largely dependent on the thermal conductivity of the base fluid. Agarwal et al. [129] synthesized water and ethylene glycol solutions containing 0–2% $Al_2O_3$ nanoparticles with different concentrations, and found that water-based nanofluids have higher thermal conductivity than ethylene glycol-based nanofluids. Conversely, the thermal conductivity of the nanoparticles is higher for ethylene glycol than for water.

*Nanoparticles:* Nanoparticles are the most important factor affecting the difference in thermal conductivity between the base fluid and the nanofluid, and their type, volume fraction, particle size and shape will all affect the lifting effect. Aberoumand et al. [130] studied the thermal conductivity of $Al_2O_3$-glycerol nanofluid through experiments, and the concentrations of nanoparticles in the fluid were 0.2%, 0.5%, 1% and 2%. The results show that, compared with the base fluid, when the volume fraction is 1%, the thermal conductivity of the nanofluid can be increased by 49%, achieving the best lifting effect. The thermal conductivity of nanofluids increases first and then decreases with the increase in the volume fraction. Li et al. [131] prepared CNT nanoparticles with concentrations varying from 0.5% to 4% using palm oil as the base fluid for grinding nickel-based alloys. The experimental results show that as the volume fraction increases, the grinding temperature first increases and then decreases, and the lowest grinding temperature is at 2%. This is consistent with the previous conclusion that the thermal conductivity of nanofluids increases first and then decreases with the increase in volume fraction. Alirezaie et al. [132] prepared 20 nm, 40 nm, 50 nm, 60 nm MgO nanofluids and 40 nm, 70 nm, 100 nm Fe nanofluids, and measured their thermal conductivities in the temperature range of 25~55 °C (at 10 °C intervals). The results show that no matter the temperature conditions, the thermal conductivity enhancement effect of 20 nm MgO and 40 nm Fe nanoparticles is the most obvious. Choi et al. [133] studied the thermal conductivity of $Al_2O_3$-water nanofluids with particle sizes of 150 nm, 47 nm, and 11 nm, and concluded that the increase in particle size reduced the thermal conductivity. Ambreen et al. [134] reviewed the effects of nanoparticles with different particle sizes on effective thermal conductivity, and found that there is an optimal particle size that improves the thermal conductivity of nanoparticles the most. Excessive particle sizes will reduce the Brownian motion speed and frequency of particles.

*Temperature:* Temperature also has an important influence on the thermal conductivity of nanofluids, and it is generally believed that a higher temperature is accompanied by higher thermal conductivity. This can be attributed to the aggressive Brownian diffusion effect, the thermal migration effect, and the higher repulsive force and lower Van der Waals force. Alirezaie et al. [132] studied the properties of MgO and Fe nanofluids at 25~55 °C and, as shown in Figure 12, the change in thermal conductivity is comprehensively affected by the type, particle size, concentration of nanoparticles and temperature. As the temperature increases, the thermal conductivity increases. Mahbubul et al. [135] synthesized $Al_2O_3$ nanoparticles in R134a, and observed the change in thermal conductivity when the temperature rose from 5 °C to 20 °C. The results showed that as the temperature Increased, the thermal conductivity increased significantly. The thermal conductivity differs by 8%. There is a non-linear positive correlation between temperature and thermal conductivity, and the strengthening trend gradually weakens. Mohanan et al. [136] measured the thermal conductivity of $Al_2O_3$-EG by the transient hot wire method at a temperature of 25 °C and 40 °C, and the experimental results obtained are consistent with the above conclusions. Suganthi et al. [137] studied the variation in the thermal conductivity of ZnO nanofluids with temperature under high-temperature conditions, and the results showed that the thermal conductivity decreased with increasing temperature. Wang et al. [37] attributed this phenomenon to the reduction in the thickness of the interference layer due to the accelerated particle motion due to the Brownian diffusion effect at higher temperatures.

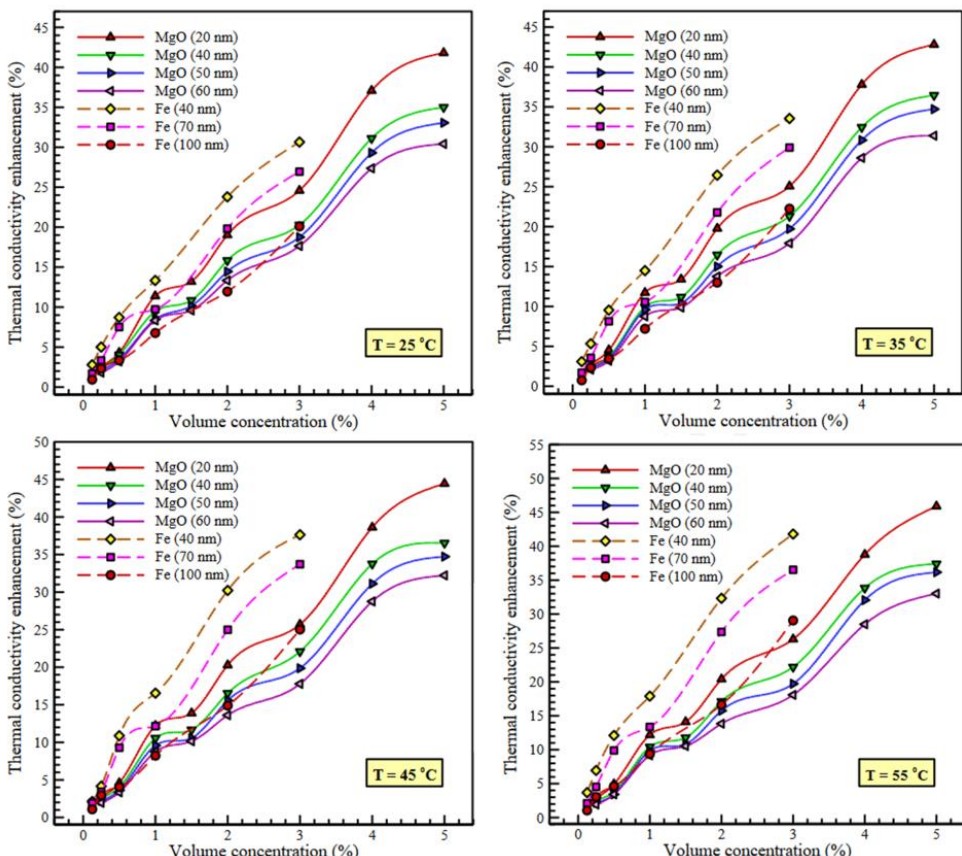

**Figure 12.** Thermal conductivity enhancement of the nanofluids Fe-EG and MgO-EG (reproduced from [132], with permission from Elsevier).

*Surfactant:* Regarding the effect of surfactants on the thermal conductivity of nanofluids, there are different opinions in different studies. Huminic et al. [138] prepared FeC-water-based nanofluids using low-viscosity carboxymethylcellulose sodium salt as a surfactant. They concluded that the surfactant does not affect the thermal conductivity of the nanofluid. Seyhan et al. [124] prepared nanofluids with gum arabic as a surfactant, Ag as nanoparticles, and water, ethylene glycol, and n-ethane as base fluids. The results showed that the gum arabic surfactant did not affect the thermal conductivity of ethylene glycol and ethane, but decreased the thermal conductivity of water-based nanofluids. Das et al. [139] added CTAB, SDBS, and SDS as surfactants to the prepared $Al_2O_3$-water nanofluids, and found that only SDBS prepared stable nanofluids, and the nanofluids containing surfactants The thermal conductivity is slightly lower than that of nanofluids without surfactants. Khairul et al. [140] interpreted this reduced phenomenon as the narrowing of the area available for heat transfer due to the surfactant. In order to explore the effect of high temperature on surfactants. Michael et al. [141] prepared BN-EG and BN-EG/DW nanofluids with PVP as the surfactant, and found that the stability decreased as the temperature increased to 60 °C. Phase separation and settling started to occur after 4 h. This is because a high temperature increases Brownian motion, reduces the viscosity of the nanofluid, and makes collisions between nanoparticles more frequent. Table 6 is the relevant research on factors affecting the thermal conductivity of nanofluids.

**Table 6.** Correlative research on influencing factors of nanofluid thermal conductivity.

| Researchers | Nanofluids | Variable | Effect |
|---|---|---|---|
| Aberoumand et al. [85] | Al$_2$O$_3$-glycerol | Concentration | As the concentration increases, the thermal conductivity first increases and then decreases, and there is an optimal concentration that maximizes the thermal conductivity. |
| Seyhan et al. [124] | Ag-water/EG/n-ethane | Surfactant, base fluid | The gum arabic surfactant did not affect the thermal conductivity of ethylene glycol and ethane, but decreased the thermal conductivity of water-based nanofluids. |
| Agarwal et al. [129] | Al$_2$O$_3$-water/EG | Base fluid | The thermal conductivity of water is higher than that of ethylene glycol, and the nanofluid prepared from water has a higher thermal conductivity. |
| Alirezaie et al. [132] | MgO/Fe | Particle size | Both nanofluids exhibit the highest thermal conductivity at the smallest particle size. |
| Choi et al. [133] | Al$_2$O$_3$-water | Particle size | The three particle sizes were 150 nm, 47 nm and 11 nm, and it was concluded that the increase in particle size decreased the thermal conductivity. |
| Mahbubul et al. [135] | Al$_2$O$_3$-R14a | Temperature | As the temperature increases, the thermal conductivity increases. |
| Huminic et al. [138] | FeC-water | Surfactant | The addition of surfactants did not have a large effect on the thermal conductivity. |

### 3.1.2. The Thermal Conductivity Model

The heat conduction model of nanofluids is a hot research field at present. According to the factors affecting the thermal conductivity of nanofluids, related models can be divided into the following types: models based on effective medium theory, models based on Brownian motion, models based on -solid–liquid boundary and models based on the aggregation model. Figure 13 shows the microscopic model of the model of nanoparticle Brownian motion, the -solid–liquid boundary layer, and aggregation effects.

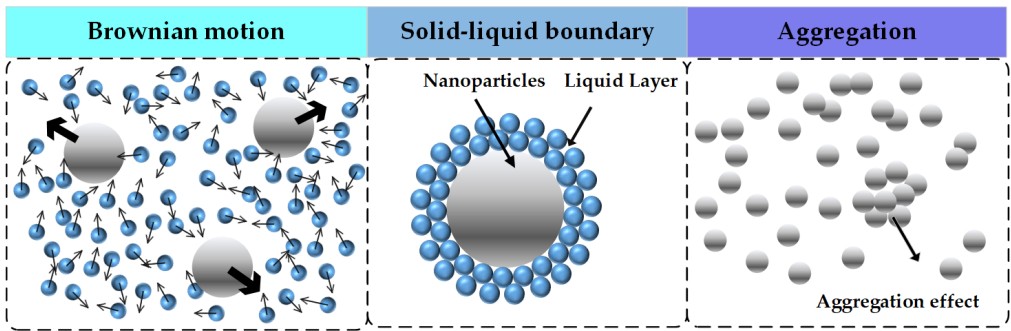

**Figure 13.** Brownian motion of the nanofluid, and the schematic diagram of the boundary layer and the aggregation effect.

The effective medium model assumes that the fluid is a homogeneous mixed medium, and uses the thermal conductivity, particle shape, and volume fraction of nanoparticles and base fluid to predict the thermal conductivity of nanofluids. Maxwell [142] added tiny solid particles to the liquid and proposed a -solid–liquid two-phase flow basic heat conduction model, but this model has limitations because he assumed spherical nanoparticles and ignored interface thermal resistance and particle aggregation. The formula is as follows:

$$\frac{K_{eff}}{K_{bf}} = \frac{K_p + 2K_{bf} + 2\phi\left(K_p - K_{bf}\right)}{K_p + 2K_{bf} - \phi\left(K_P - K_{bf}\right)} \tag{1}$$

where $K$ represents thermal conductivity, subscript *eff* represents mixed fluid, *bf* represents the base fluid, *p* represents particles, and *f* is particle concentration.

In order to determine the effect of the thermal conductivity of the nanoparticle suspension, Hamilton and Crosser [143] considered the difference in particle shape and introduced the particle shape factor n to propose a H-C thermal conductivity calculation model. The formula is as follows:

$$\frac{K_{eff}}{K_{bf}} = \left[ \frac{K_p + (n-1)K_{bf} - (n-1)\phi\left(K_p - K_{bf}\right)}{K_p + (n-1)K_{bf} + \phi(K_p - K_{bf})} \right] \tag{2}$$

This formula is valid for both cylindrical and spherical nanoparticles, where *n* is the shape factor, $n = 3/y$, and y is the sphericity.

Bruggeman et al. [144] considered the osmotic effect of nanoparticles and explained the asymmetry of the mixture. The formula is as follows:

$$\frac{K_{eff}}{K_{bf}} = \frac{1}{4}\left[(3\phi - 1)K_p + (2 - 3\phi)K_{bf}\right] + \frac{K_{bf}}{4}\sqrt{\Delta} \tag{3}$$

where $\Delta = \varphi^m$, it is the correction factor of heterogeneity and microstructure. $\varphi$ is the volume fraction of the discontinuous phase. *m* is Bruggeman coefficient. The Bruggeman coefficient depends on the microstructure and properties of the complex medium; when $m = 1$, the model degenerates to the classical model. The equal-volume average thermal conductivity model is suitable for some uniformly distributed systems. When $m = 2$, the model is suitable for the spherical surrounding system, in which the discontinuous phase surrounds the continuous phase in a spherical shape. The value of *m* may take different ranges between 1 and 2. These are the classic effective medium theory models.

The Brownian motion model examines how the random motion of nanoparticles within the base fluid influences the thermal conductivity of nanofluids. Keblinski et al. [145] theoretically analyzed the Brownian motion of nanoparticles at low and medium temperatures. At low and moderate temperatures, the motion's time scale exceeds that of thermal diffusion in the host fluid by over two-fold, resulting in a negligible contribution to direct heat transfer. Evans et al. [146] observed that the aggregation effect of nanoparticles produces micro-convection in the surrounding liquid, thereby indirectly transferring heat, and Brownian motion plays a somewhat lower role in the heat transfer than expected. In summary, the Brownian motion of nanoparticles is not the main mechanism leading to the increase in the thermal conductivity of nanofluids.

According to the boundary layer model, nanoparticles possess a semi-solid or liquid crystal thin layer on their surface, resulting in interfacial thermal resistance at the interface between different components of the mixture, a phenomenon referred to as Kapitza interfacial resistance. Yu and Choi [147] demonstrated that layered molecules exist in a physical state between solids and fluids. The solid-like nanolayer composed of liquid molecules exhibits higher thermal conductivity than the underlying liquid. This signifies that the solid-like nanolayer functions as a pivotal solid bridge between the nanoparticles and the liquid, playing a crucial role in augmenting thermal conductivity. The formula is as follows:

$$\frac{K_{eff}}{K_{bf}} = 1 + \frac{\phi K_p}{1 - K_{bf}} A \tag{4}$$

where *A* is the correction parameter.

Yu and Choi [148] modified the Maxwell model to take into account the effect of the boundary layer on the effective thermal conductivity of the fluid. They postulated that the nanolayers enveloping each particle could amalgamate with the particles, resulting in the formation of equivalent composite particles. Due to the model's confinement to spherical nanoparticles, the researchers extended the Hamilton—Crosser model to encompass non-



spherical nanoparticles. This adaptation considered the impact of nanolayers. The formula is as follows:

$$\frac{K_{eff}}{K_{bf}} = \frac{K_p + 2K_{bf} + 2\phi\left(K_p - K_{bff}\right)(1+\beta)^3}{K_p + 2K_{bf} - \phi\left(K_p - K_{bf}\right)(1+\beta)} \tag{5}$$

Here, β is the relation between nanolayer thickness and particle diameter.

The aggregation effect model considers the potential formation of nanoparticles into variously sized and shaped aggregates within the base fluid. This alteration influences the effective volume fraction and shape factor of the nanofluid, subsequently impacting its thermal conductivity. Xuan et al. [149] provided a model considering Brownian motion and nanofluid cluster aggregation effects. This model is unique, but ignores the influence of interfacial thermal resistance on thermal conductivity. The formula is as follows:

$$\frac{K_{eff}}{K_{bf}} = \frac{K_p + 2K_{bff} + 2\phi\left(K_p - K_{bf}\right)}{K_p + 2K_{bf} - \phi\left(K_p - K_{bf}\right)} + \frac{\phi\rho_p C_p}{2K_{bf}}\sqrt{\frac{2K_b T}{3\pi r\mu}} \tag{6}$$

Here, *r* is the cluster diameter, $K_b$ is Boltzmann constant, and $\mu$ is the viscosity.

### 3.2. Viscosity

Viscosity signifies the flow resistance within nanofluids. During the cutting process, higher nanofluid viscosity results in increased resistance, impeding cutting fluid replenishment in the processing region [150]. This situation also diminishes the heat convection coefficient due to a heightened nanofluid boundary layer caused by viscosity increment. Simultaneously, nanofluid viscosity profoundly impacts the film-forming and lubricating abilities of the cutting fluid within the processing domain [151]. Consequently, the viscosity of nanofluid-based cutting fluids should not be pursued indiscriminately to achieve either excessively high or excessively low values.

### 3.2.1. Influencing Factors

*Nanoparticles:* The shape and size of nanoparticles are crucial parameters that influence the viscosity of nanofluids. Minakov et al. [152] systematically measure the viscosities of over 30 distinct nanofluids. Nanoparticle sizes ranged from 5 to 150 nm. The findings revealed that nanofluid viscosity is largely contingent on nanoparticle material, and viscosity increases as nanoparticle size decreases. However, Hu et al. [153] arrived at a contrary conclusion. They observed that as the particle size decreases, the viscosity also decreases. They analyzed this discrepancy and attributed it to varying nanoparticle concentrations. They identified three correlations between viscosity and particle size: ① a positive correlation, ② a constant trend, ③ a negative correlation. Ferrouillat et al. [154] employed Ag to synthesize spherical and "banana-shaped" $SiO_2$ and ZnO nanoparticles for the formulation of water-based nanofluids. The correlation between dynamic viscosity and shape exhibits minimal influence. The impact of particle shape and size on nanofluid viscosity presents conflicting influences, necessitating additional systematic investigations to yield definitive conclusions. Koca et al. [155] reviewed the effect of particle size on the viscosity of nanofluids, many studies have limited particle size ranges and varied experimental conditions. Summarizing a specific theoretical relationship to elucidate the particle size—viscosity connection is challenging due to this variability. Figure 14 takes $Al_2O_3$-water nanofluid as an example to show the combined effects of particle size, concentration, and temperature on viscosity.

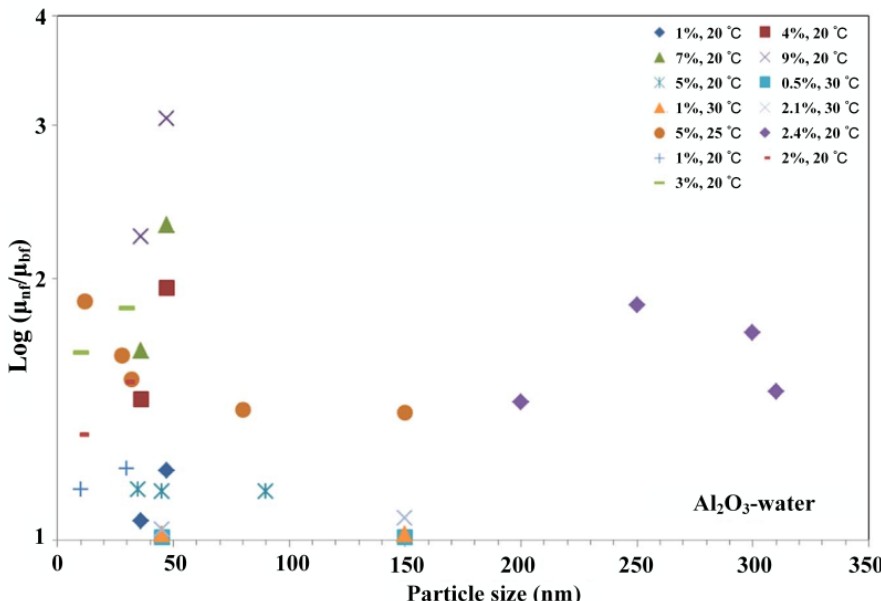

**Figure 14.** Relationship between relative viscosity ($\mu_{nf}/\mu_{bf}$) and particle size of Al$_2$O$_3$-water nanofluids in the temperature range of 20~30 °C (reproduced from [155], with permission from Elsevier).

*Temperature:* Viscosity tends to decrease as temperature increases and thermal conductivity increases. Jabbari et al. [156] conducted an experiment involving single-walled carbon nanotube-water nanofluids placed in a temperature range of 25~65 °C, observing the change in its viscosity. Their experimental findings demonstrated that as the nanofluid's temperature decreased, its viscosity increased. Similarly, lyahraja et al. [157] observed a comparable outcome when investigating Ag nanofluids, revealing a non-linear negative correlation between temperature and viscosity. Bindu et al. [136] examined viscosity changes in a ternary nanofluid composed of "Al$_2$O$_3$-ZnO-MWCNT-EG". The experimental data indicated that the viscosity index at 70 °C was only approximately half of that at 40 °C, highlighting a substantial downward trend. Table 7 is related research on factors affecting the viscosity of nanofluids.

**Table 7.** Correlative research on influencing factors of nanofluid viscosity.

| Researchers | Nanofluids | Variable | Effect |
|---|---|---|---|
| Bindu et al. [142] | Al$_2$O$_3$-ZnO-MWCNT-EG | Temperature | The viscosity at 70 °C was only approximately half of that at 40 °C, highlighting a substantial downward trend. |
| Minakov et al. [152] | More than 30 kinds | Particle size | Viscosity increases as nanoparticle size decreases. |
| Hu et al. [153] | Al$_2$O$_3$/ZnO | Particle size | As the particle size decreases, the viscosity also decreases. |
| Ferrouillat et al. [154] | SiO$_2$/ZnO | Particle shape | The impact of particle shape and size on nanofluid viscosity presents conflicting influences. |
| Jabbari et al. [156] | SWCNT-water | Temperature | As the nanofluid's temperature decreased, its viscosity increased. |
| lyahraja et al. [157] | Ag nanofluid | Temperature | Temperature and viscosity have a non-linear positive correlation. |

### 3.2.2. The Viscosity Prediction Model

Einstein's model is the most widely used classical viscosity prediction model, which derived from the linear fluid dynamics equation [158]. This formula only uses the viscosity coefficient of the base fluid and the particle concentration to estimate the viscosity coefficient

of the suspension, and is suitable for dilute suspensions of spherical particles. The specific formula expression is as follows:

$$\frac{\mu_{eff}}{\mu_{bf}} = 1 + 2.5\phi \tag{7}$$

Among them, $\mu$ represents the viscosity, the angle subscript *eff* represents the suspension, *bf* represents the base fluid, and $\phi$ represents the concentration of the suspension.

However, this equation is limited to predicting the viscosity behavior of spherical rigid particles and low-concentration fluids. Subsequently, Brinkman [159] formulated a viscosity prediction equation applicable to high concentrations, based on Einstein's equation.

$$\frac{\mu_{eff}}{\mu_{bf}} = 1 - \frac{\phi}{\phi_m} \tag{8}$$

$\phi_m$ is the relative maximum packing density.

Batchlor [160] further enhanced the equation by considering the impact of Brownian motion on viscosity. The formula is as follows:

$$\frac{\mu_{eff}}{\mu_{bf}} = 1 + 2.5\phi + 6.5\phi^2 \tag{9}$$

Brenner and Condiff developed a new viscosity prediction model that taking into account the shape effect of rod-like particles in dilute suspensions [161]. This model is defined for viscosity at high shear rates and is suitable for prediction of suspension viscosity up to $1/r^2$ volume fraction. Its specific formula is as follows:

$$\frac{\mu_{eff}}{\mu_{bf}} = 1 + \eta\phi \tag{10}$$

Here, $\eta = \frac{0.312r}{\ln 2r - 1.5} + 2 - \frac{0.5}{\ln 2 - 1.5} - \frac{1.872}{r}$ and $r$ is the aspect ratio of the particles.

Many models are based on conventional fluid flow viscosities, but they often lack the necessary precision to predict the viscosities of nanofluids. However, with the increasing research and application of nanofluids, the prediction model of nanofluid viscosity has received widespread attention.

Noni et al. [162] proposed a parametric viscosity model suitable for ceramic nanoparticle suspensions, taking into account particle concentration and two-dimensional force balance. The formula is as follows:

$$\frac{\mu_{eff}}{\mu_{bf}} = 1 + b\left(\frac{\phi}{1 - \phi_m}\right)^n \tag{11}$$

Here, $b$ and $n$ are constants and can be calculated by least squares regression.

Tseng and Lin [163] considered that particle clusters are an important factor affecting the rheological behavior and suspension structure of nanofluids. They considered clusters as an important factor in the viscosity model and proposed a new viscosity model for water-based TiO$_2$ nanofluids. The formula is as follows:

$$\frac{\mu_{eff}}{\mu_{bf}} = \left(1 - \frac{\phi}{0.605}\left(\frac{a_a}{a}\right)^{1.2}\right)^{-1.5125} \tag{12}$$

Here, $a_a$ and $a$ are the diameter of aggregates and primary particles, respectively.

Murshid et al. [164] introduced empirical correlations to establish a new viscosity-related prediction model for silicone oil-based peninsular nanofluids. The formula is as follows:

$$\mu_{eff} = \mu_0\left(A + Be^{-CT}\right) \tag{13}$$

Here, $\mu_0$ is the reference viscosity of base fluids at room temperature. $T$ is temperature, and $A$ is the dimensionless empirical coefficients. $B$ and $C$ in the correlation are functions of nanoparticle types and concentration.

### 3.3. The Specific Heat Capacity

The specific heat capacity indicates the heat storage capacity of a substance. Precisely estimating this capacity is vital for characterizing nanofluids' heat transfer performance in real-world applications [83]. Hassan et al. [165] measured the specific heat capacities of $Al_2O_3$ nanofluids with volume fractions of 0.5, 1, 1.5, 2 and 2.5% at different temperatures. The base fluid used in the experiments was mixed with different ratios of glycol and water. The research shows that the specific heat capacity decreases with the increase of the proportion of ethylene glycol, because the specific heat capacity of ethylene glycol is smaller than that of water, which indicates that the specific heat capacity of the nanofluid is positively correlated with that of the base fluid. Noraldeen et al. [166] systematically studied the effect of particle size and interface temperature on the specific heat capacity of nanofluids. The experimental results showed that the specific heat capacity increased with the increase of nanoparticle size, but decreased with the increase in temperature, the changing trend of which is shown in Figure 15. This situation is attributed to interfacial effects.

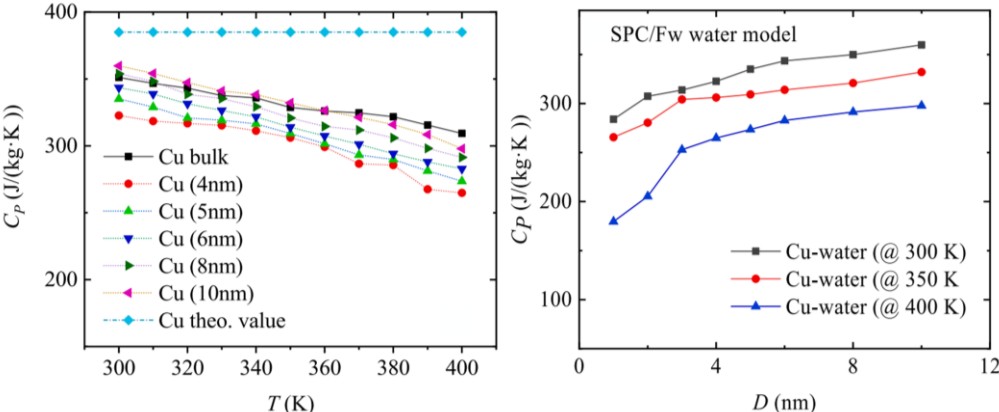

**Figure 15.** Variation trend of specific heat capacity with temperature and particle size (reproduced from [166], with permission from Elsevier).

Currently, researchers employ theoretical models grounded in ideal gas mixing theory and heat balance principles for determining the specific heat capacity of nanofluids at varying concentrations. The formula is as follows:

$$C_{pnf} = \frac{(1-\phi)\rho_{bf}C_{pbf} + \phi(\rho C_P)\rho_P C_{pp}}{(1-\phi)\rho_{bf} + \phi\rho_P} \tag{14}$$

Among them, $f$ represents the volume fraction, $\rho_{bf}$ represents the density of the base fluid, $\rho_P$ represents the density of the particles, $C_{pbf}$ represents the specific heat capacity of the base liquid, and $C_{pp}$ represents the specific heat capacity of the particles.

Buongiorno used the above model to calculate the specific heat capacity of nanofluids based on the concept of heat capacity [167], assuming that the nanoparticles and the base fluid are in thermal equilibrium, while Alammar and Hu calculated the specific heat capacity of nanofluids based on the mixing theory of ideal gas mixtures using the equation [168]:

$$C_{Pnf} = (1-\phi)(C_P)_{bf} + \phi(C_P)_P \tag{15}$$

In order to measure the specific heat capacity of mixed nanofluids, Seawram proposed an artificial neural network (ANN) model [169]:

$$C_{pnf} = \frac{Q}{m_{nf}\Delta T} \tag{16}$$

where $Q$ is heating transfer rate, $m_{nf}$ is mass flow rate of the hybrid nanofluid, and $\Delta T$ is temperature difference.

The addition of nanoparticles significantly improves the thermal conductivity of the cutting fluid, reduces the thermal damage of the parts during processing, and improves the surface integrity of the parts. In addition, as the particle concentration increases, the dynamic viscosity of the fluid will also increase, which will affect the fluidity of the droplet after it enters the tool- interface, slowing down the replacement of cutting fluid, and the accumulation of chips and heat is not conducive to workpiece processing. However, at the same time, the higher viscosity value is beneficial to ensuring stable lubrication within the working range. It is necessary to select an appropriate concentration to balance the influence of viscosity on workpiece processing.

## 4. The Nano-Cutting Fluid Cooling and Lubrication Mechanism

Nano-cutting fluid is not directly applied to flood processing, but as the atomization medium of MQL, with a small amount of cutting fluid, it can achieve the same level or even better cooling and lubrication effects as flood cooling [170]. As a solid additive, nanoparticles make the cooling and lubricating method of nano-cutting fluids very different from traditional cutting fluids, in which nanoparticles play a unique role [171]. This section will explain the cooling and lubrication mechanism of the nano-cutting fluid from the three aspects of the anti-friction and anti-wear mechanism, the cooling and lubrication mechanism and the atomization penetration mechanism endowed by MQL.

### 4.1. The Anti-Friction and Anti-Wear Mechanism

In cutting machining, the unique mechanism of action of nanoparticles at the -tool–workpiece interface has attracted extensive attention from researchers [172–174]. Through induction and summary, the following four modes of action are obtained. (1). Nanoparticles form a bearing-like structure at the processing interface, which converts the sliding friction between the tool and the workpiece into rolling friction, effectively reducing the friction coefficient between the two working surfaces, especially spherical nanoparticles, which are well adapted to the nanoparticle "rolling effect". (2). In layered solid lubricating materials such as GR, the shear force between the nano-layers is relatively weak—it can be easily peeled off during the sliding process, and then adheres to the working surface to form a dense and smooth solid friction film, further reducing friction and wear. (3). The nanoparticles in the cutting fluid will move to the surface defects of the workpiece under the action of external force factors. The instantaneous high temperature during the sliding process can melt nanoparticles with low melting points, repair the micro-damage on the surface of the workpiece, make the surface of the workpiece smooth, and avoid further damage to the surface integrity of the workpiece and damage to the knives. (4). For nanoparticles made of high-hardness materials such as ND, the nanoparticles themselves have a good polishing effect, reduce the roughness of the friction pair, reduce the contact surface stress, and improve the bearing capacity of the cutting fluid. This is called the polishing effect. The mechanism of this effect is shown in Figure 16.

Force, tool wear, and surface roughness are all cutting parameters that can intuitively reflect the reduction in -tool–workpiece interface friction.

*Force*: In the turning process of AA6061-T6, the use of nanofluids as cutting fluids can reduce the feed force, the radial force and the tangential force in the turning process by 13.46%, 16.23% and 6.34% respectively. Under the mechanism of reducing wear and anti-friction of nanoparticles, the cutting force in the cutting process has been significantly reduced [173].

*Tool wear*: In the cutting process, the adhesion of materials and the corrosion of cutting fluids are the main causes of tool wear. During NMQL machining, nanoparticles changed the friction mechanism between the -tool–workpiece interface, and the tool wear state was also significantly improved. Turning EN24 steel with NMQL reduces tool wear by 75% compared to dry machining [175].

*Surface roughness*: Surface roughness is defined as the shorter frequency of the actual surface compared to the trough, and it is the most intuitive parameter used to evaluate the processing quality of parts. While improving the friction mode, nanoparticles' unique filling effect can greatly improve the quality of the processed surface, thereby reducing the surface roughness. In the AISI 321 drilling process, NMQL reduced the surface roughness by 33.8% [176].

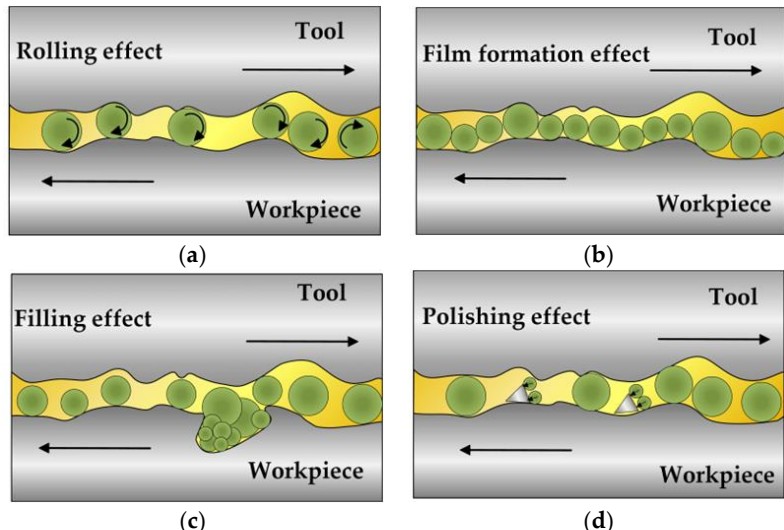

**Figure 16.** The anti-friction and anti-wear mechanism of nanoparticles: (**a**) the rolling effect; (**b**) the film formation effect; (**c**) the filling effect; (**d**) the polishing effect.

The relevant studies listed in Table 8 use cutting parameters to characterize the anti-friction and anti-wear effects of nanoparticles.

**Table 8.** Evaluation of cutting parameters on anti-friction and anti-wear effect of nanofluids.

| Researchers | Nano-Cutting Fluid | Characterization Parameters | Effect |
|---|---|---|---|
| Rapeti et al. [173] | NPI-coconut oil | Flank wear | The contributions of base fluid and nanoparticles to reducing flank wear are 46.6% and 26.18%, respectively. |
| Naresh et al. [175] | Cu nanofluids | Surface roughness, flank wear | The surface roughness and flank wear of Cu nanofluid + MQL have been improved by 88% and 75%, respectively, which is 9% higher than that of casting nanofluids. |
| Pal et al. [176] | NG-vegetable oil | Flank wear, Cutting force, surface roughness, | The cutting force is reduced by 27.4%, the torque is reduced by 64.9%, the surface roughness is reduced by 33.8%. |
| Anandan et al. [177] | Ag nanofluids | Flank wear, Cutting force | 60% reduction in flank wear and 18% reduction in cutting force. |
| Rapeti et al. [178] | NPI-coconut oil | Cutting force | The contribution of base fluid and nanoparticles to the reduction in cutting force is 72.86% and 8.6%, respectively. |

**Table 8.** *Cont.*

| Researchers | Nano-Cutting Fluid | Characterization Parameters | Effect |
|---|---|---|---|
| Rapeti et al. [178] | NPI-coconut oil | Surface roughness | The contribution of base fluid and nanoparticles to the reduction in surface roughness is 22.5% and 19.2%, respectively. |
| Yi et al. [179] | GR oxide nanofluids | Cutting force, surface roughness | 17.21% reduction in cutting force and 15.1% reduction in surface roughness. |
| Malladi et al. [180] | Boric acid/ coconut oil | Cutting force | Compared with pouring machining, the cutting force decreases by 12.74% under NMQL working conditions. |
| Pashmforoush et al. [181] | Cu/water | Surface roughness | Cu-water nanofluid improves the surface roughness of the workpiece by 62.16%. |
| Khajehzadeh et al. [182] | TiO$_2$/water | Flank wear | Tool flank wear is affected by particle size, with an increase in the size of nanoparticles from 10 to 50 nm, the decrease flank wear reduces from 46.2% to 34.8%. |

*4.2. The Enhanced Heat Transfer Mechanism*

The addition of nano-additives to the base fluid further enhances the thermal conductivity and convective heat transfer capabilities of the base fluid, as solid substances generally have better cooling properties than liquids. Tiara et al. [183] used Al$_2$O$_3$ nanofluid jet AISI304 steel plate to detect the change in heat flux, and the detection of surface roughness proved that the number of active sites increased, which proved that the addition of nanoparticles improved heat transfer efficiency. Singh et al. [184] used three different nanofluids of TiO$_2$, Al$_2$O$_3$, and SiO$_2$ to impact the heating plate to detect the change in heat flux. The results showed that compared with water, the three nanofluids significantly improved the change in heat flux. Among them, the enhancement effect of TiO$_2$ nanofluid is the most obvious. The addition of nanoparticles improves the heat transfer performance of the base fluid through two mechanisms. One is that the high surface energy of the nanoparticles adsorbs liquid molecules to form a fluid adsorption layer, which effectively reduces the thermal resistance between the particles and the liquid. The second is that Brownian motion forces the nanoparticles to hit the workpiece, so heat is transferred from the workpiece to the nanoparticles. At the same time, when the nanoparticles move randomly and collide with each other, heat channels will be formed between the adsorption layers, and the excess heat will be effectively taken away from the cutting area. The general heat transfer mechanism of nanoparticles in base fluid is shown in Figure 17.

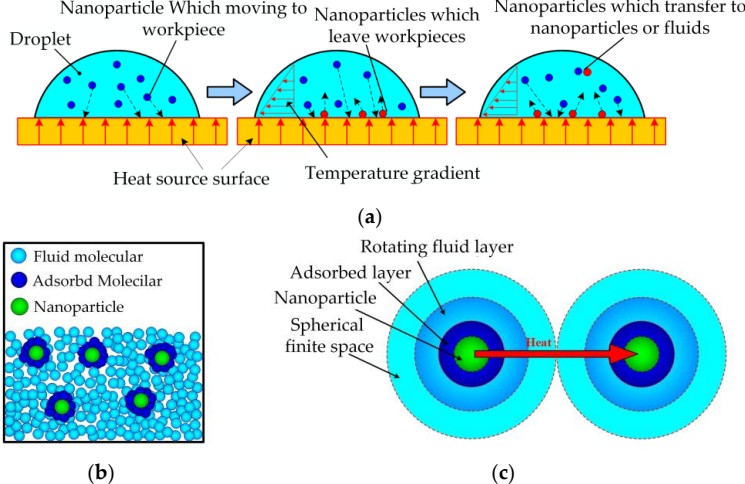

**Figure 17.** Heat transfer mechanism of nanoparticles: (**a**) heat transfer process; (**b**) adsorption layer of nanoparticles; (**c**) heat transfer channels [6]. Copyright (2022), with permission from Springer Nature.

*Temperature*: Cutting temperature is the parameter that can most directly characterize the heat transfer in the cutting process [67]. If the heat released instantaneously during the material removal process cannot be discharged in time, it may cause thermal cracks in the workpiece and cause thermal damage. Nanoparticles have been well documented in improving thermophysical properties. Table 9 shows methods to improve the heat transfer of nanofluids as cutting fluids in the cutting process.

**Table 9.** Evaluation of cutting parameters on the enhanced heat transfer effect of nanofluids.

| Researchers | Nano-Cutting Fluid | Characterization Parameters | Effect |
|---|---|---|---|
| Naresh Babu et al. [175] | Cu-EG | Cutting temperature | Cutting temperature reduced by 53%. |
| Anandan et al. [177] | Ag nanofluids | Cutting temperature | Under the optimal parameters, the cutting temperature is reduced by 44%. |
| Rapeti et al. [178] | NPI-coconut oil | Cutting temperature | The cutting temperature decreases, the base fluid and nanoparticles contribute 67% and 13.77%, respectively. |
| Yi et al. [179] | GR oxide nanofluids | Thermal cracks | GR oxide nanofluid is used to reduce thermal cracks by 15.1%. |
| Safiei et al. [185] | $SiO_2$-$Al_2O_3$-$ZrO_2$ nanofluids | Cutting temperature | The cutting temperature of NMQL end milling aluminum alloy is reduced by 76% and 16%. |
| Yi et al. [186] | GR oxidenanofluids | Cutting temperature | Compared with the traditional cutting fluid, the cutting temperature decreased by 27.16 °C, 30.42 °C, and 31.8 °C. |
| Li et al. [187] | GR/LB2000 | Cutting temperature | The milling temperatures of pure MQL and GR MQL decreased by 33.62% and 67.64%, respectively. |
| Zetty Akhtar et al. [188] | $TiO_2$, $Al_2O_3$/EG | Cutting temperature | Both nanoparticles significantly reduced the cutting temperature and $TiO_2$ has a better performance improvement effect. |

*4.3. The Atomization and Penetration Mechanism*

The atomization process means that in the device, the cutting fluid is atomized into a gas-liquid mixture through high pressure and gas, and then introduced into the cutting area through the nozzle jet [189]. The atomized medium delivered by high pressure can more effectively pass through the air barrier layer formed by the high-speed rotation of the cutter. The small-sized droplets produced by the atomization process have lower surface tension, which makes them have a smaller contact angle after attaching to the processed surface [190]. It can be more widely distributed on the surface of the workpiece, thereby more effectively absorbing and taking away the heat on the surface of the workpiece, and forming a lubricating film to reduce friction. In addition, small droplets penetrate the capillary structure of the workpiece more quickly [191]. Compared with traditional wet cutting, MQL reduces the atomization process at high cutting temperatures, and the penetration efficiency is significantly improved. Figure 18a is a schematic diagram of the structure of high-surface-energy and low-surface-energy droplets after atomization. Figure 18b shows the state of droplets with different surface tensions spreading on the surface of the workpiece. Additionally, Figure 18c shows a schematic diagram of the droplet penetration mechanism.

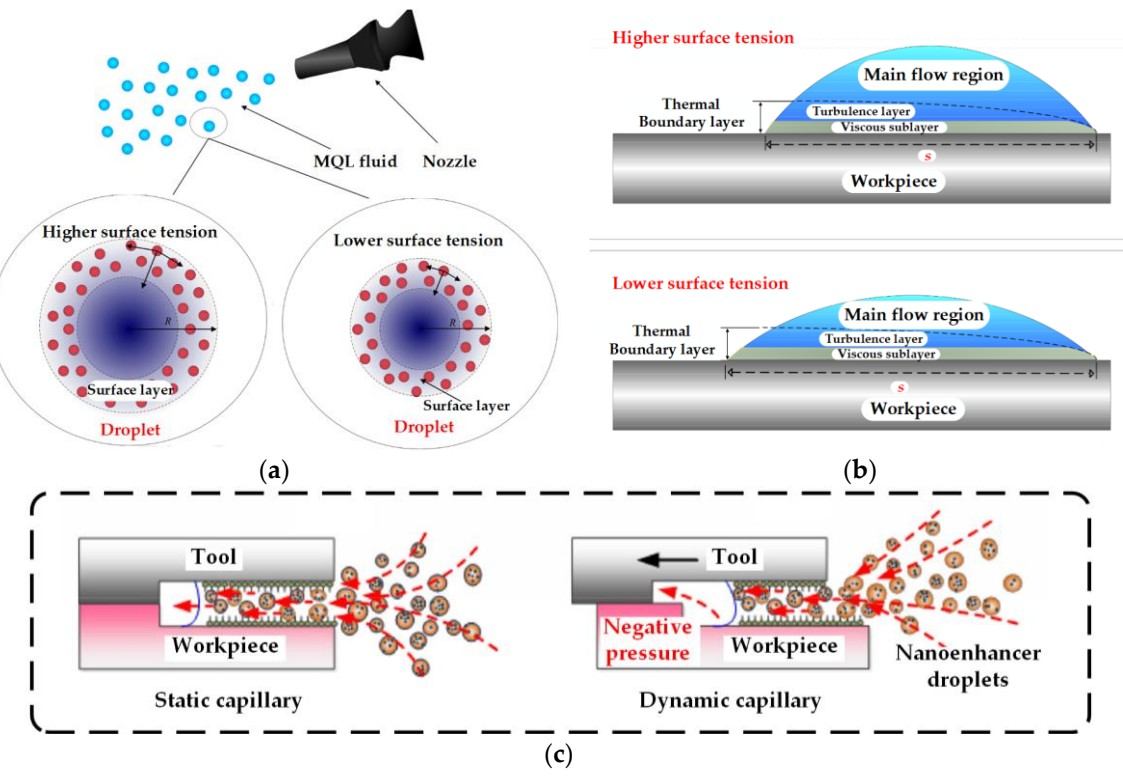

**Figure 18.** (**a**) Atomization mechanism and droplet surface tension; (**b**) influence mechanism of contact angle (reproduced from [192], with permission from Elsevier); (**c**) the penetration mechanism [6]. Copyright (2022), with permission from Springer Nature.

In extreme processing environments, insufficient penetration performance caused by NMQL's poor transmission energy is a challenge when cutting difficult-to-process materials. To address this problem, the researchers chose to add a magnetic field and use a magnetic nanofluid as the cutting fluid. Fluids can move in a controlled manner under the influence of magnetic fields to enhance the fluid's permeability [193]. Zhang et al. [194] used a $Fe_3O_4$ magnetic nanofluid with a magnetic field to conduct cutting experiments. The results of the cutting experiments showed that at the maximum magnetic field intensity, the cutting force was reduced by 36.9% and the surface roughness was reduced by 28.15%. Jiao et al. [195] use magnetic abrasive particles to remove ring groove material under the action of a magnetic field. The surface roughness dropped from the initial *Ra* 4.3 "μm" to *Ra* 0.6 "μm". Zhang et al. [196] detected that when using magnetic nanofluids, compared with conventional cutting fluids, the cutting force was reduced by 48.6% and the surface roughness of the machined workpiece was reduced by 49.1% in turning experiments. Cui et al. [197] verified the feasibility of magnetic traction nanofluid Ti-6Al-4V grinding and revealed its unique penetration and friction reduction mechanism. In this study, the grinding force and grinding temperature were reduced by 35.8% and 66.4%, respectively, and the arithmetic mean height was reduced by *Sa* by 27.5%. In NMQL without a magnetic field, the driving force of the nanofluid to the grinding zone is only the viscous force between the nanofluid and the grinding wheel. The magnetic field will exert an additional adsorption force on the magnetic nano-cutting fluid, which is much higher than the viscosity force and gravity. The oil film is tightly adsorbed on the surface of the grinding wheel and enters the grinding area smoothly. The penetration mechanism of magnetic nanofluid is shown in Figure 19a. In addition, under the influence of a magnetic field, magnetic nanoparticles will be regularly arranged along the direction of the magnetic field to form a chain structure. The chain-like arrangement of nanoparticles and the state under relative motion between the tool and the workpiece are shown in Figure 19b. The regularly arranged

chain structure is conducive to the film-forming effect of nanoparticles, thereby reducing the grinding force and further improving the surface quality.

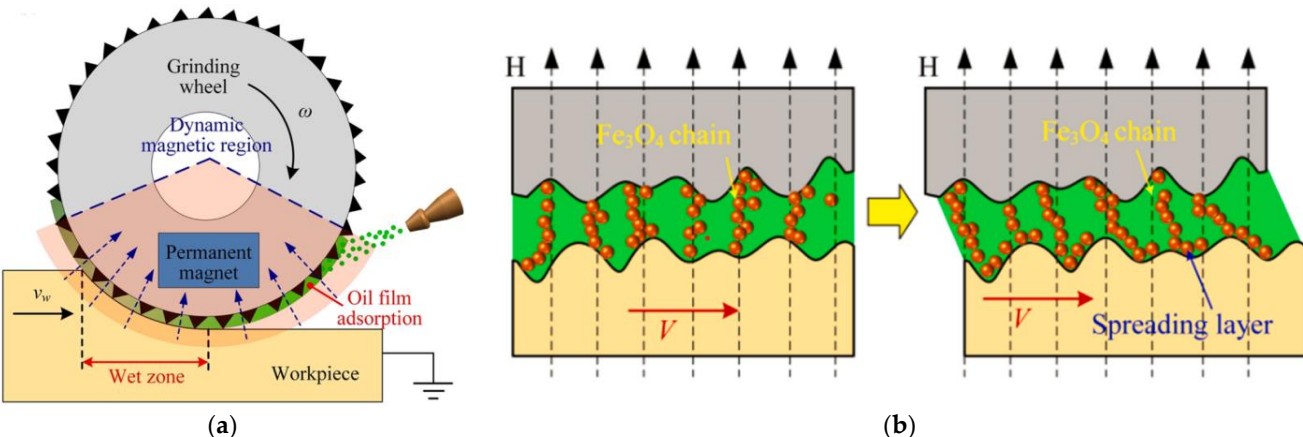

**Figure 19.** Magnetic nanofluid under the influence of magnetic field: (**a**) the penetration mechanism; (**b**) chain arrangement of nanoparticles (reproduced from [197], with permission from Elsevier).

### 4.4. Dialectical Perspectives

Through a dialectical analysis of nanofluid characterization, improvements in nanofluid properties largely depend on nanoparticles. However, the interaction between the base fluid and the nanoparticles will affect the rheological behavior of the nanofluid, including shear refinement and shear thickening, thus affecting the pumping power and flow characteristics in practical applications. In the application of nano-cutting fluids, the Brownian motion of nanoparticles in the base fluid and the heat exchange of the -solid–liquid thermal boundary layer are the bases for the enhanced heat transfer of nanofluids. Likewise, the Brownian motion of nanoparticles affects their dispersion and behavior in the base fluid, affecting van der Waals forces to a certain extent, thus counteracting precipitation.

Nano-cutting fluids required for industrial applications are usually prepared using a two-step process. After selecting a specific type of nanoparticles, it is difficult to use the particle shape to influence the machining performance of the prepared cutting fluid. The fluid concentration is an important parameter that affects the thermal conductivity, viscosity, and specific heat capacity of nanoflow. Additionally, it is the easiest particle index to control during the preparation process. Therefore, selecting the optimal particle concentration and base fluid mixing ratio determines the practicality of the prepared nano-cutting fluid. Table 10 shows relevant research on the optimal concentration of the nano-cutting fluid.

**Table 10.** Assessment of the optimal particle concentration of nano-cutting fluids.

| Researchers | Nanofluid/ Based Fluid | Concentration | Working Conditions | Experimental Results |
|---|---|---|---|---|
| Ngoc et al. [198] | $MoS_2$/ soybean oil | 0.2%, 0.5%, 0.8% | Turning 90CrSi | Nanofluid at a concentration of 0.5% has the greatest impact on turning forces. |
| Singh et al. [199] | Graphite-talc/ coconut oil | 0.25%, 0.75%, 1.25%. | Turning Titanium alloy | The nanofluid with a concentration of 0.75% has the lowest friction coefficient and the best surface quality after processing. |
| Zhang et al. [200] | $MoS_2$-CNT/synthetic lipids | 2%, 4%, 6%, 8%, 10%, 12% | Grinding GH4169 Ni-based alloy | Combined friction coefficient and *Ra*, 8% $MoS_2$-CNTs was the optimum concentration for nanofluid MQL. |

Table 10. *Cont.*

| Researchers | Nanofluid/ Based Fluid | Concentration | Working Conditions | Experimental Results |
|---|---|---|---|---|
| Li et al. [201] | GR/LB2000 | 0.05 wt.%, 0.1 wt.%, 0.15 wt.%, and 0.2 wt.%. | Grinding TC4 alloy | The minimum grinding temperature peak (183.612 °C) is obtained when the concentration of cutting fluid is 0.1 wt.% |
| Singh et al. [202] | alumina-GR/ water | 0.25%, 0.75%, 1.25%. | Turning AISI 304 | 1.25% hybrid nanofluid showed minimum wear rate. |

Note: Red represents the optimal concentration.

## 5. Evaluation of the Processing Performance of Different Materials

NMQL has emerged as a promising technology in machining processes. NMQL enhances machining performance and aligns with sustainability goals [203]. Given the rising need for efficient and environmentally friendly processing practices, assessing the performance of nanofluidic MQLs becomes vital, particularly across various material types. This section is to thoroughly explore the effectiveness of NMQL in different cutting environments.

### 5.1. Common Materials

AISI 1045 steel, a medium carbon steel, finds common application in machining processes and the fabrication of machine components, gears, and shafts. Baldin et al. [204] added 0.05 wt.% and 1 wt.% GR nanosheets to mineral-based oil to make nanofluids when milling AISI 1045 steel, and used dry cutting, vegetable oils as controls, respectively. The results showed that the nano-based plant cutting fluid effectively improved the lubricating performance, but the tool life was lower than that of the experimental tool under pure vegetable oil cutting fluid, which may be because GR failed to dissolve completely in the base fluid. The diameter of the aggregated particle clusters is too large, which intensifies the abrasive wear mechanism. Abbas et al. [205] compared dry cutting, flood cutting, and nanofluid cutting based on minimum lubrication, and turned AISI 1045 steel in three ways. The results show that the surface quality of machining with nanofluid as cutting fluid is significantly better than that of the other two groups, and the consumption of NMQL is the least in terms of power consumption. The only downside is that the cost consumption is higher than dry cutting. Huang et al. [206] demonstrated that using MWCNT nanofluid as a cutting fluid during high-speed milling of AISI 1045 steel can significantly reduce the cutting temperature, achieve better surface finish, reduce flank wear, and increase tool life. Ni et al. [207] used three different nanoparticles, $Fe_3O_4$, $Al_2O_3$, and carbon, as nano-additives to improve the cutting performance of vegetable oil cutting fluids in Broaching AISI 1045 steel, and combined with commercial cutting fluids (CCF) as cutting fluids. For comparison, the results showed that the cutting fluid containing carbon nanoparticles produced the best lubrication effect. This cutting fluid reduced the average broaching load peak and load valley to 725 N and 614 N, which were 115.7% and 118.5% higher than those of CCF; and its root mean square error of broaching vibration signal was 36.15% lower than CCF.

Due to their high strength-to-weight ratio, good corrosion resistance, and weldability, aluminum alloys are often used as a substitute for low-carbon steel in engineering applications and are widely used in aerospace, automotive components, and structural applications [208]. However, due to their strong adhesion, aluminum alloys tend to stick to the tool material during machining, which shortens the tool life and affects the surface integrity of the part [209]. As an emerging processing method, NMQL has been widely recognized because it plays an important role in improving processing performance. Taking aluminum alloy 6061-T6 as an example, the performance improvement of using nanofluid as a cutting fluid in the cutting process of aluminum alloy under different processing conditions was discussed. Najiha et al. [210] used $TiO_2$ nanofluid as cutting fluid to study the wear mechanism of the rear surface of the tool during end milling of AA6061. A nanofluid

improves the integrity of the workpiece surface by improving microwear and adhesion during machining. At the same time, in terms of cooling speed, compared with traditional oil-based cutting, water-based nanofluids perform better. In another paper, Najiha et al. [79] compared the effect of $TiO_2$ particle concentration on processing conditions. At a nanoparticle concentration of 0.5%, insufficient lubrication resulted in chip buildup, the machined flank shown in Figure 20. However, too high a concentration (4.5%) also failed to achieve the desired effect. Flank wear is at the lowest and edge integrity is at the highest the concentration is 2.5%. Niknam et al. [211] established a model for MQL turning AA6061-T6 and AA7076-T6, and experimentally proved that there is a close relationship between cutting fluid viscosity and surface roughness. The high viscosity of nano-cutting fluids is positively correlated with the surface integrity of parts after machining. In addition, the higher the flow rate of the nano-lubricant, the better the surface integrity of the part. This is because the faster flow rate removes chips and heat more quickly, reducing surface burn and chip scoring. Insufficient cooling properties of the cutting fluid can lead to heat buildup. This may exacerbate sticking during cutting with AA7076-T6 in high-temperature environments. To solve this problem, Safiei et al. [185] used $SiO_2$-$Al_2O_3$-$ZrO_2$ ternary nanofluid as the cutting fluid, which was applied to milling AA6061-T6 material. The nanofluid mist provided by the MQL method helps to overcome the high-speed gas barrier and forms an adhesive film at the cutting interface, thereby minimizing the interaction and preventing chips from adhering to the cutting edge. During the cutting process, only a small amount of nanofluid needs to be delivered to reduce the cutting temperature at the interface. Figure 21 shows the synergistic effect of cutting parameters and MQL parameters, as well as cutting parameters and fluid parameters on the cutting temperature of AA6061-T6.

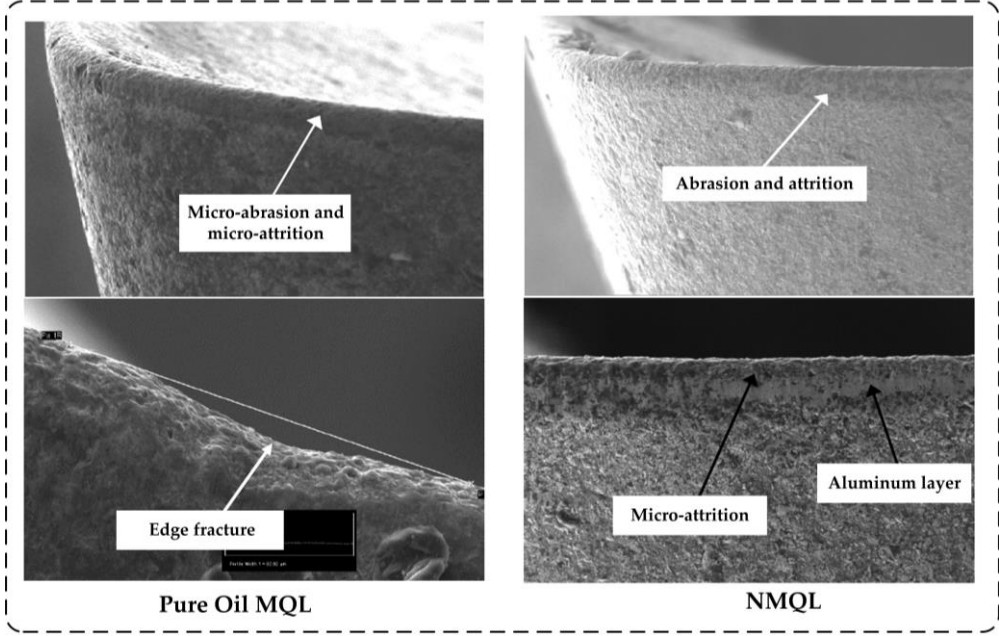

**Figure 20.** SME of flank wear and edge wear after aluminum alloy milling (reproduced from [79], with permission from Elsevier).

The excellent corrosion resistance and hygienic properties of stainless steel make it the first choice for kitchen utensils, food processing equipment and medical equipment [212]. Stainless steel is used on building facades, handrails and roofs for its aesthetics and structural strength. In addition, stainless steel is a common material in the manufacture of automotive components, industrial machinery and transportation systems due to its excellent mechanical properties of high strength and durability [213]. Similarly, factors such as high temperature and built-up edge in the traditional cutting process seriously affect the productivity of stainless steel. Researchers began to turn their attention to the optimization

process of nanofluid micro-quantity lubrication [214]. Akincioğlu et al. [215] studied the effect of a new green hexagonal boron nitride nanofluid on the friction coefficient of AISI 316L stainless steel. The experimental verification results showed that the friction coefficient was reduced by 77.64% compared with dry cutting. Ukamanal et al. [216] used water-based $Al_2O_3$ nanofluid as the cutting fluid in the turning process of AISI 316 stainless steel. The experimental verification results showed that when nanofluid micro-cutting fluid was used, the tool temperature was the lowest, thereby avoiding the thermal deformation of the tool and the workpiece. Saatci et al. [217] discussed the sustainable optimized processing of AISI 310 austenitic stainless steel, using $MoS_2$ nanofluid micro-quantity lubrication as a new processing method. The experimental results show that this method reduces the surface roughness by 86.8%, carbon emissions were reduced by 30.4%, and processing costs were also relatively reduced by 2.9%. Pal et al. [176] used pure vegetable oil and CNT/Vegetable oil nanofluid as cutting fluid when drilling AISI 321 stainless steel with minimal lubrication. The experimental results showed that under the condition of the nanofluid, thrust, torque, surface roughness and the friction force was reduced by 27.4%, 64.9%, 33.8% and 51.7%, respectively. In addition, Pal et al. [218] also used $Al_2O_3$ nanofluid as a cutting fluid for AISI 321 drilling, and compared it with dry machining and flood machining. They concluded that nanofluids produced better results in MQL drilling, due to the cooling-lubricating properties of the nanoparticles, resulting in a higher cooling effect. Maruda et al. [219] analyzed the effect of nanoparticle size and concentration on turning AISI 316L stainless steel to further optimize nanofluids as cutting fluids and lubricants. As shown in Figure 22, when the nanoparticle size is 22 nm and the concentration is 0.5%, the most regular peak-to-valley variation is observed. It can be observed that as the size of the nanoparticles increases, the difference in the variation in the surface peaks and valleys also increases. This is because size has a more significant effect on vibration during cutting.

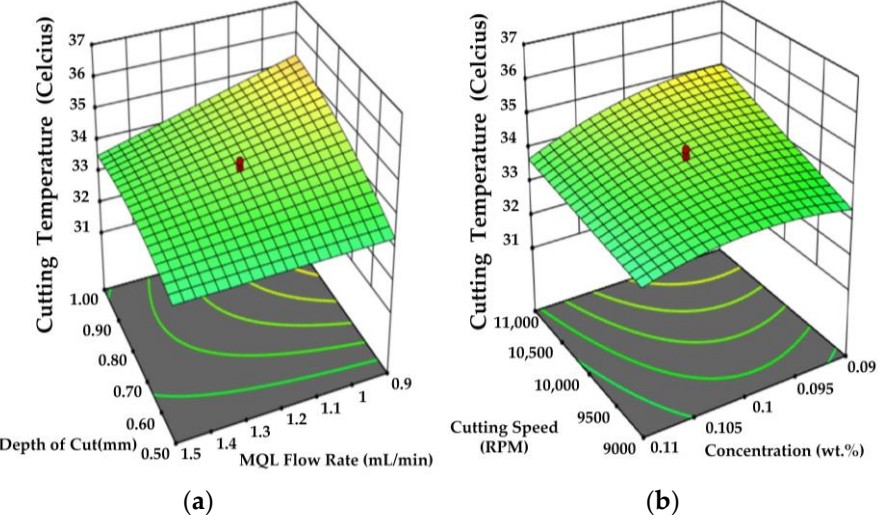

(**a**)              (**b**)

**Figure 21.** The 3D graphical plots of the cutting temperature under the interaction of two factors: (**a**) cutting speed and concentration; (**b**) depth of cut and MQL flow rate (reproduced from [185], with permission from Springer Nature).

### 5.2. Difficult-to-Machine Materials

Modern technology has more and more higher requirements for the efficiency and technology of workpiece production [219]. However, conventional machining techniques face formidable challenges when dealing with difficult to machine materials due to their unique properties and behavior [220]. Machining these materials often encounters challenges such as high flank wear, poor surface finish, work hardening, thermal deformation, and a low machinability index [221]. This section will discuss how nanofluids effectively overcome these challenges in material processing.

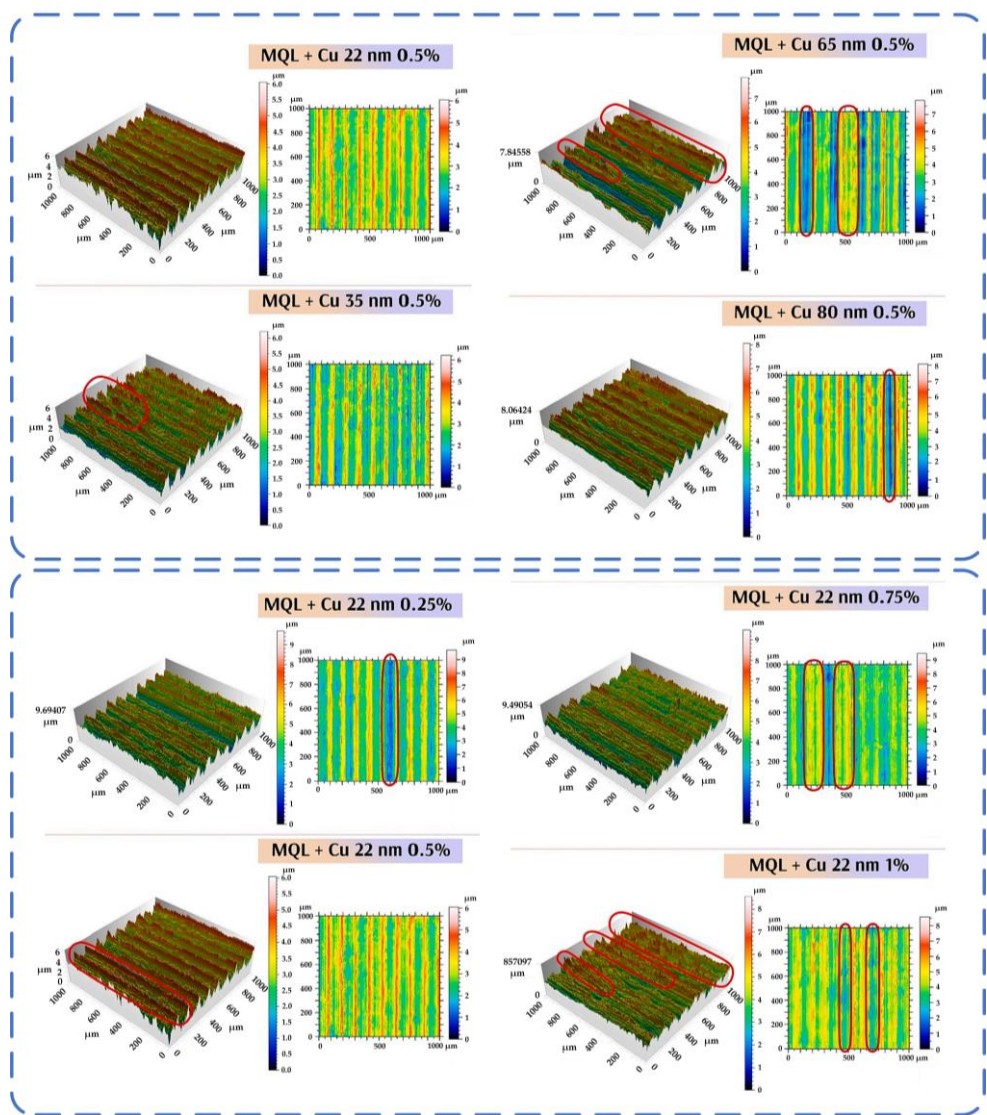

**Figure 22.** Surface morphology of AISI 316 after turning under different conditions (reproduced from [219], with permission from Elsevier).

### 5.2.1. Titanium Alloys

Titanium alloys exhibit a medium density, a high strength-to-weight ratio, excellent oxidation resistance, a low coefficient of thermal expansion, high toughness, and good weldability, making them widely applicable in aerospace, biomedicine, and automotive manufacturing [222]. Friction at the tool—chip interface and material adhesion represent the primary challenges in titanium alloy machining. Ti-6Al-4V stands as a typical example of a challenging-to-machine titanium alloy material. Ti-6Al-4V's high strength to weight ratio and low thermal conductivity result in a rapid increase in cutting temperature during the machining process, leading to the risk of workpiece surface burns and flank wear [223]. Gupta et al. [224] used liquid nitrogen supplemented with minimal quantity lubrication to achieve a sustainable cryogenic machining environment. They chose to use CVD-coated tools for turning Ti-6Al-4V. Under such a machining environment, they successfully achieved the smooth surface quality after titanium alloy machining, while minimizing chip accumulation and flank wear.

In the cutting of titanium alloys, conventional cutting fluids are difficult to effectively lubricate. Nano-cutting fluids not only the lubricating performance can be improved, but also the problem of thermal conductivity of titanium alloy can be solved to a certain extent.

Yi et al. [179] used GR oxide suspension as cutting fluid during Ti-6Al-4V drilling process. The results show that compared with the traditional cutting fluid, the GR oxide suspension cutting fluid reduces the cutting temperature by 58%, the surface roughness decreases by 17.21%, and the chip is thinner. The addition of GR oxide significantly improves the difficulty of titanium alloy cutting Phenomenon. Zhou et al. [225] carried out laser micro-grooving treatment on uncoated tools and used nanofluid as cutting fluid in their research on milling titanium alloy Ti-6Al-4V. The improvement effect of cutting parameters is verified through experimental comparison with traditional cutting tools and cutting fluid. The results showed that under the improved cutting conditions, the cutting force was reduced by 38.4%, the surface roughness was reduced by 27.75%, and the flank wear rate was reduced by 63.3%. In addition, the researchers observed a mechanism for the coupling effect of textured cutters and nanofluids. Bai et al. [226] conducted nanofluid micro-lubrication milling experiments on Ti-6Al-4V titanium alloys. In the experiment, six different types of nanofluids were used as cutting fluids to evaluate their impact on the improvement of titanium alloy milling performance. The experimental results show that the milling force of the $Al_2O_3$ nanofluid group exhibits a minimum value. In terms of surface roughness, the value of $Al_2O_3$ is also the smallest. The surface quality and specific experimental data are shown in Figure 23. $Al_2O_3$ and $SiO_2$ nanoparticles are considered to be more suitable as environmentally friendly additives for base oils in the processing of titanium alloy metals. Gaurav et al. [227] evaluated the effect of two different $MoS_2$ nanofluids (one based on jojoba oil and the other based on LRT30) on the machining performance of turning Ti-6Al-4V. The experimental results show that the performance improvement effect of $MoS_2$ nanoparticles on jojoba oil is better than that on LRT30 mineral oil, especially in reducing the cutting force during turning, and also in improving surface roughness and flank wear. Better effect. Anandan et al. [177] improved the turning environment of titanium alloys by adding Ag nanoparticles to the cutting fluid. Under the optimal parameter conditions, the surface roughness was reduced by 89%, and the turning temperature was reduced by 44%.

### 5.2.2. Nickel-Based Alloys

Nickel-based alloys occupy an important position in the engineering field due to their unique mechanical, thermal and chemical properties. Due to its excellent high-temperature performance, it is often used in gas turbines, jet engines and other aerospace components [51]. In addition, its excellent corrosion resistance makes it particularly suitable for chemical and marine environments. However, its high strength, hardness and thermal conductivity lead to serious flank wear during cutting, and it is difficult to dissipate heat in the cutting zone. Under high pressure stress, it also causes severe plastic deformation, which induces work hardening and surface roughness [228]. To improve the machining performance of nickel-based alloys, nanofluid micro-lubrication has become a promising method.

Li et al. [131] evaluated the effect of different concentrations (0.5–4%) of CNTs nanofluids on the grinding performance of nickel-based alloys. The experimental results show that 2% volume fraction of nanofluid exhibits the best lubrication and heat transfer performance in the grinding process. The cutting force is 21.93 N, 21.30 N under the condition of 2.5% volume fraction, and the cutting temperature is 108.9 °C, which is the lowest temperature in the control group. Similarly, Huang et al. [228] also studied the effect of different nanoparticle types on the machinability of Ni-based alloys. They selected six different kinds of nanoparticles as variables and the grinding temperature as a control condition. Ultimately, they found that $SiO_2$ nanoparticles are more suitable as nano-additives in the grinding process of nickel-based alloys. Sen et al. [229] mixed $Al_2O_3$ with palm oil to form a nanofluid and applied it in MQL milling of Inconel-690. This significantly reduces specific cutting energy and flank wear during milling. The conclusions show that the MQL nanofluidic synergy can easily replace conventional lubrication technologies to alleviate economic and environmental challenges. Venkatesan et al. [230] noticed that the Inconel 617

material was very hard in the dry turning environment, and further used $Al_2O_3$ nanofluid micro-lubrication to improve the cutting environment, and analyzed the flank wear by fitting the regression model and concluded that the nanofluid greatly improved Inconel 617 material is difficult to process. Şirin et al. [231] set up a variety of different cutting lubrication conditions to test the milling performance of Inconel X-750 superalloy. The experimental results show that surface roughness, the cutting temperature, the cutting force, flank wear and tool life are the most performance characteristics. The best results are obtained under nanofluid cutting conditions, and the optimum concentration is 0.5 vol%. Zhang et al. [232] carried out nanofluid MQL grinding experiments, and used $Al_2O_3/SiC$ without a mixing ratio as the cutting fluid to conduct grinding experiments on nickel-based alloys. The experimental results show that the size and mixing ratio of nanoparticles will significantly affect the final grinding effect.

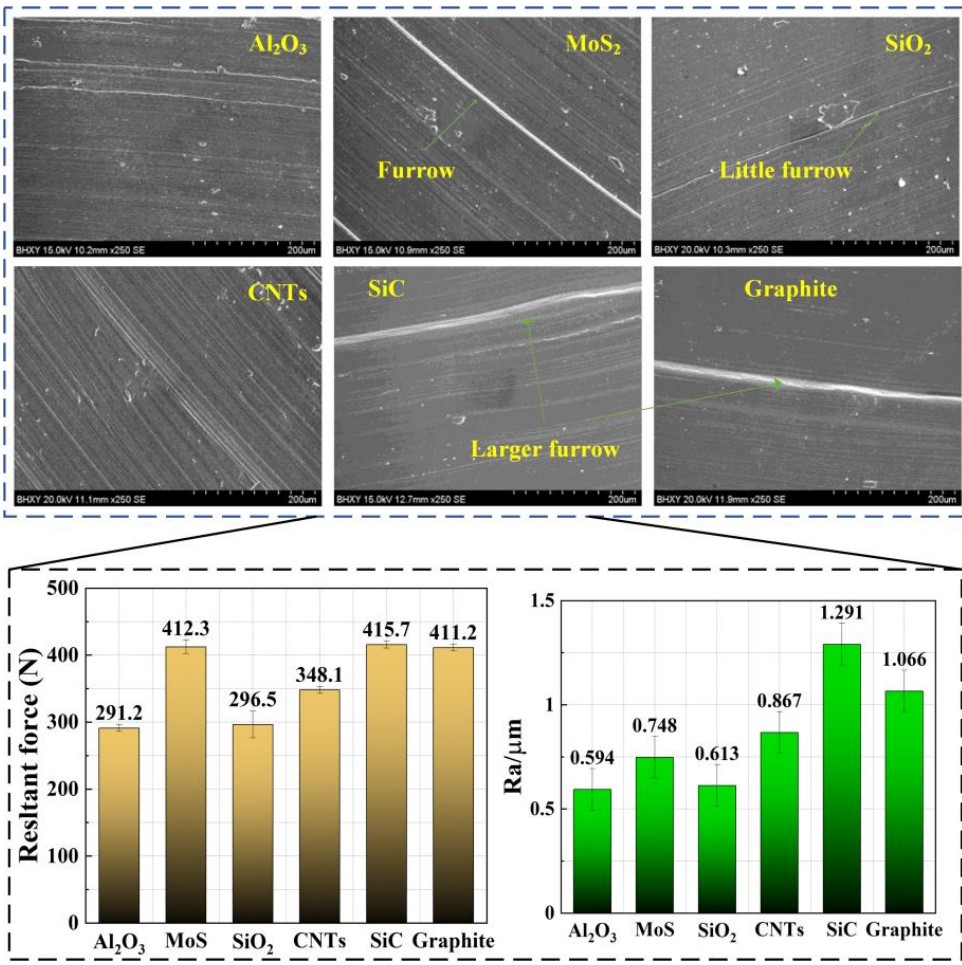

**Figure 23.** SME of surface quality after milling Ti-6V-4A, and its resultant force and *Ra* (reproduced from [226], with permission from Springer Nature).

### 5.2.3. Hardened Steel

AISI 4140 is a common low alloy quenched steel known for its high strength, hardness and good toughness. AISI 4140 is widely used in the manufacture of key components such as shafts, gears, and bearings, and is also used in the aerospace field [233]. However, due to its relatively high carbon content and the martensite formed during quenching, machining and forming becomes challenging and can lead to rapid flank wear. Elbah et al. [234] chose to use coated hybrid ceramic tools to process AISI 4140 steel with high carbon content and obtained good processing results. Furthermore, AISI 4140 steel is very sensitive to heat treatment. During cutting, an excellent cooling method is essential to achieve the mechanical properties required for AISI 4140. Researchers have applied NQML as an effective

cooling measure in the machining of AISI 4140 steel. This section will take AISI 4140 as a typical quenched steel to review the influence of nanofluids on cutting performance.

Roy et al. [235] observed that nanofluids showed better heat absorption characteristics compared with traditional cooling lubrication in high-speed turning of AISI 4140 steel. Sayuti et al. [236] used $SiO_2$ nanofluid as cutting fluid to study the optimal lubrication parameters in the AISI 4140 hard turning process, and used logic and response analysis to confirm the effective lubricity of nanofluids. Gürbüz et al. [237] evaluated the sustainability of the dry, wet and MQL cutting methods in AISI 4140 steel cutting. The experimental results and mathematical model analysis showed that MQL is superior to dry and wet processing in terms of sustainability and cleaner production. Furthermore, Gürbüz et al. [238] also pointed out that in the MQL cutting of AISI 4140 steel, the size of the flow rate will affect the final processing result, and verified the influence of the size of the flow rate on the cutting force during the machining process. Muthuvel et al. [239] evaluated drilling experiments in AISI 4140 steel in a Cu NMQL environment. It was found that due to the good thermal conductivity of Cu nanoparticles, the high temperature in the cutting zone was successfully eliminated and the flank wear could be effectively reduced. In addition, nanofluids have excellent penetration capabilities that minimize the formation of jaggies. Khajehzadeh et al. [240] used $TiO_2$ nanofluid in hard turning AISI 4140 to improve heat transfer capability and tribological properties, and obtained the best cutting parameters through simulation and experimental verification. When the concentration of nanofluid was 3.0 wt.%, the tool and cutting contact area length reduced by 35%.

*5.3. Composite Material*

Composite materials are defined as the combination of two or more synergistic micro-components, including two components of matrix and reinforcement, with three-dimensional domains of specific characteristics between these components [241]. The composite material is composed of a reinforcement phase and a matrix phase, which can simultaneously exert the excellent properties of the two materials. The matrix is usually a metal, polymer or ceramic material, while the reinforcement is mainly fibers, particles and crystalline filaments [242]. Due to their heterogeneous structure, composite materials are considered difficult to machine, and using traditional cutting methods usually leads to failure phenomena such as material matrix fragmentation, fiber extraction, swelling and delamination [243]. It is an important condition for the application of reinforced composite materials to choose an appropriate processing method to improve the cutting environment and reduce the scrap rate of materials. It has been proved that NMQL is an effective cooling and lubrication method in cutting processing, and researchers have also widely used this method in cutting processing of composite materials to verify the feasibility of its processing [244]. This section will review the improvement effect of nanofluids on the machining environment in the cutting process of two typical composite materials, CFRP and metal matrix composites (MMC).

CFRP are highly compatible, and the use of composite materials in fuselage design is shown in Figure 24 [245]. Such composites are typically manufactured by near-net shape, often using precision machining in cutting processes such as grinding and milling. Delamination, burrs and tears are common problems during the processing of carbon fiber-reinforced composites [246]. Different from metal materials, the CFRP used in composite materials has hygroscopicity, which causes the CFRP material to absorb water and expand, which seriously affects its mechanical properties, thus limiting the feasibility of flood cooling and lubrication in cutting processing [247]. Dry machining often results in a poor surface quality, tool damage and high rates of workpiece damage. Cooling and lubricating properties are one of the key factors in processing CFRP materials [248]. Researchers have begun to apply NMQL as an effective cooling and lubrication method in the cutting process of CFRP materials.

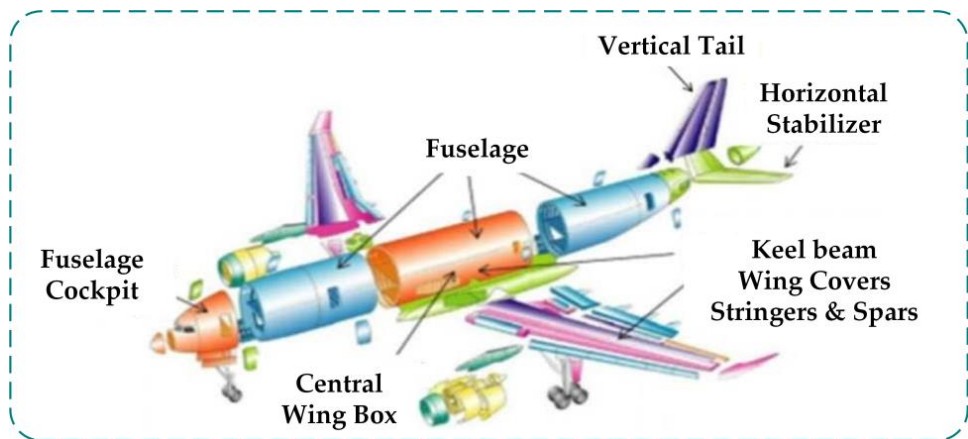

**Figure 24.** Large-size CFRP composite components used in Airbus 350 (reproduced from [245], with permission from Springer Nature).

Nicholas et al. [249] showed that the application of MQL technology in the CFRP machining process can improve the surface finish of the workpiece, reduce flank wear, and significantly reduce the burr phenomenon in the machining process. Gao et al. [250] verified the superiority of MQL used in the CFRP milling process. The oil film formed by MQL is between the milling cutter and the surface of the workpiece, which not only significantly reduces the cutting force and surface roughness, but also effectively reduces the common burrs in CFRP processing. Pradana et al. [251] selected nanoparticles as nano-additives added to distilled water for milling of CFRP. The experimental results show that under the selected optimal parameters, the $TiO_2$/water nanofluid cooling medium has the most significant cooling effect on the cutting temperature, which can greatly reduce the flank wear of the tool. Qu et al. [252] studied the improvement effect of carbon nanofluids on the grinding performance of carbon fiber-reinforced ceramic matrix with minimal quantity lubrication, and verified the positive effect of NMQL on reducing the grinding force and improving surface roughness through experiments. The cutting force and surface quality data under four different lubrication conditions are shown in Figure 25. It can be seen intuitively that there is an improvement effect of NMQL on composite material cutting. In addition, Qu et al. [253] studied the effects of nozzle orientation, air pressure, oil flow rate and nozzle distance on the MQL grinding performance of unidirectional Cf/SiC composites, and determined the optimal parameter. Mughal et al. [254] used nanofluid micro-lubrication to improve flank wear during CFPR/Ti-6Al-4V composite machining, and verified through experiments that the addition of $MoS_2$ nanoparticles improved flank wear, the wear was positively correlated with ion concentration, and the concentration increased from 0.5% to 1%; flank wear is also reduced from 85 μm to 75 μm.

Metal matrix composites (MMCs), consisting of ductile metals reinforced with ductile particles, fibers, or whiskers are widely used in aerospace due to their high strength-to-weight ratio, low cost, and processability [255]. However, due to their anisotropy and heterogeneous structure, MMC materials are often considered difficult to process. Aluminum-based metal composites are favored by researchers because of their good economy. However, in the process of processing MMC, especially under the traditional processing technology, the material scrap rate is high. In order to solve this problem, researchers prefer to use fine processing methods such as milling and grinding to process MMC [256]. At the same time, as a process that can significantly improve the cutting performance, MQL has been introduced into the research of cutting MMC. As a commonly used medium for improving cooling performance of MQL, nanofluids have also attracted the attention of some researchers.

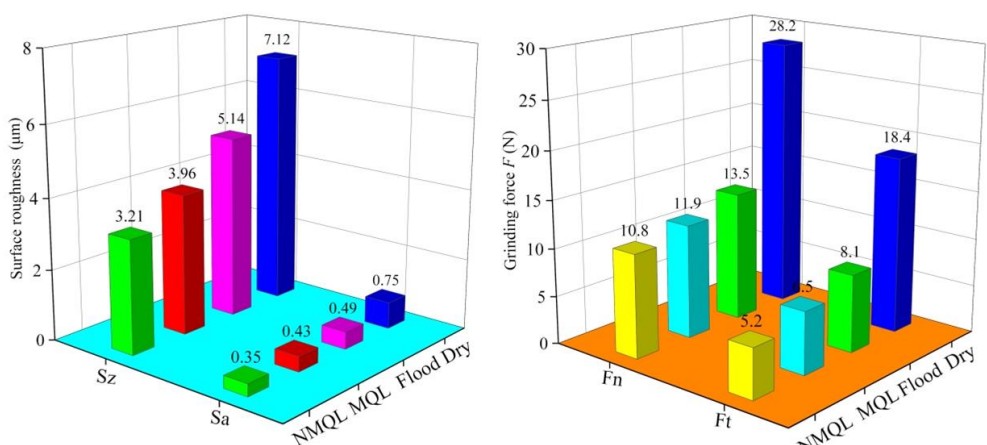

**Figure 25.** Composite machining under different lubrication conditions: surface roughness and grinding force (reproduced from [252], with permission from Elsevier).

Davim et al. [257] studied the influence of the cutting fluid flow rate on machining quality during MQL machining of MMC. Experiments prove that MQL can significantly improve surface roughness. An increase in flow rate reduces surface roughness, although this effect is weak. James et al. [258] developed the AA6061-ZrO$_2$ composite metal matrix material and performed MQL turning on the new material. The experiment proves that the flow rate of cooling and lubricating mist is the main factor affecting the surface roughness and flank wear, while the depth of cut is the main factor affecting the cutting force. Kilickap et al. [259] studied the optimal cutting parameters of steel matrix composites in MQL drilling. Experimental results show that MQL can achieve better surface roughness compared to dry machining and low temperature machining. Vishwas et al. [260] studied the surface roughness results of cutting Al/SiC metal matrix composites by adding SiC through stirring during the fabrication of aluminum matrix composites. Under NMQL conditions, the cutting parameters that achieve the best surface quality are determined. Nandakumar et al. [261] further introduced nanofluids as cutting fluids on the basis of MQL cutting of Al/SiC MMCS, which significantly reduced the temperature of the processing area during cutting. Chakma et al. [262] optimized the turning machining of aluminum matrix composites using carbon nanotube nanofluids. The experimental results show that the nanofluid-assisted tool can achieve higher cutting speeds and lower feed rates, thereby significantly improving the surface quality. Sujith et al. [263] added Al$_2$O$_3$ to pure coconut oil for cutting Al-7079/7 wt.%-TiC in situ reinforced metal matrix composites and evaluated its processing properties. The experimental results show that the addition of suitable amounts of nanoparticles can improve the cutting performance of cutting fluids, but excessive nanoparticles will lead to a significant increase in cutting force, flank wear, surface roughness and the cutting temperature.

The important published experimental studies of nano-cutting fluids on machining of different materials were compared horizontally. Studies have consistently shown that nano-cutting fluids can provide good cutting performance for the processing difficulties of various materials, whether it is the problem of easy adhesion of aluminum alloy or the problem of difficult cutting of cemented carbide. Nanofluids used in experimental studies must be well characterized in terms of particle size, size distribution, shape and aggregation for the broadest applicability of results. Once the science and engineering of nanofluids are fully understood and their full potential investigated, they can be replicated on a large scale and used in many applications.

*5.4. Sustainability Assessment in NMQL*

With the increasing awareness of environmental protection among people, the concept of sustainability in the field of metal cutting has attracted widespread attention from re-

searchers. Sustainable production places new demands on reducing energy use, pollution, emissions and health risks. In terms of specific energy, compared with dry, flood and MQL, NMQL cutting can be reduced by 40–42%, 25–30% and 17–19%, respectively, at relatively low combined friction and heat flux density. From an energy perspective, NMQL is an efficient and sustainable processing method, and nanofluids have become the first choice for sustainable production due to their high efficiency and environmental protection. Deiab et al. [264] evaluated the energy consumption during turning Ti-6Al-4V with six different lubrication methods. Experimental results show that NQML with vegetable oil-based nanofluid as cutting fluid consumes the least energy. However, energy consumption is not an adequate proxy for sustainability assessment. Khan et al. [265], based on energy consumption, added factors such as surface integrity, tool life, production cost and assigned weighted proportions to determine the overall performance index (OPI) to evaluate the sustainability of different lubrication cooling methods. Research results show that due to the difficulty and hazards of nanofluid preparation, the OPI of NMQL is slightly lower than that of MQL, but this slight impact can be compensated for by adopting appropriate safety and environmentally friendly procedures and following waste management standard instructions. Ic et al. [266] further used the carbon emissions generated during the processing stage as an influencing factor to develop a sustainability assessment model. Yi et al. [267] not only considered the impact of carbon emissions, but also considered the environmental burden caused by the use of resources.

Singh et al. [268] investigated the performance of Hastelloy C-276 sustainability assessment in different processing environments. The results showed that MQL has 9–27% lower energy consumption and carbon emissions compared to dry and cast methods, as well as 9–24%. Moreover, since the amount of cutting fluid used by MQL is much less than that of pouring lubrication, it also has great advantages in terms of costs in cleaning, recycling, and disposal of waste cutting fluid. As shown in Figure 26, Padhan et al. [269] conducted a sustainability assessment of the dry cutting method and the NMQL cutting method after considering many factors such as surface finish, score, and environmental impact. The data comparison of the clearance ring chart can be seen. NQML is superior to dry processing in many aspects, and using NMQL processing has economic and social benefits.

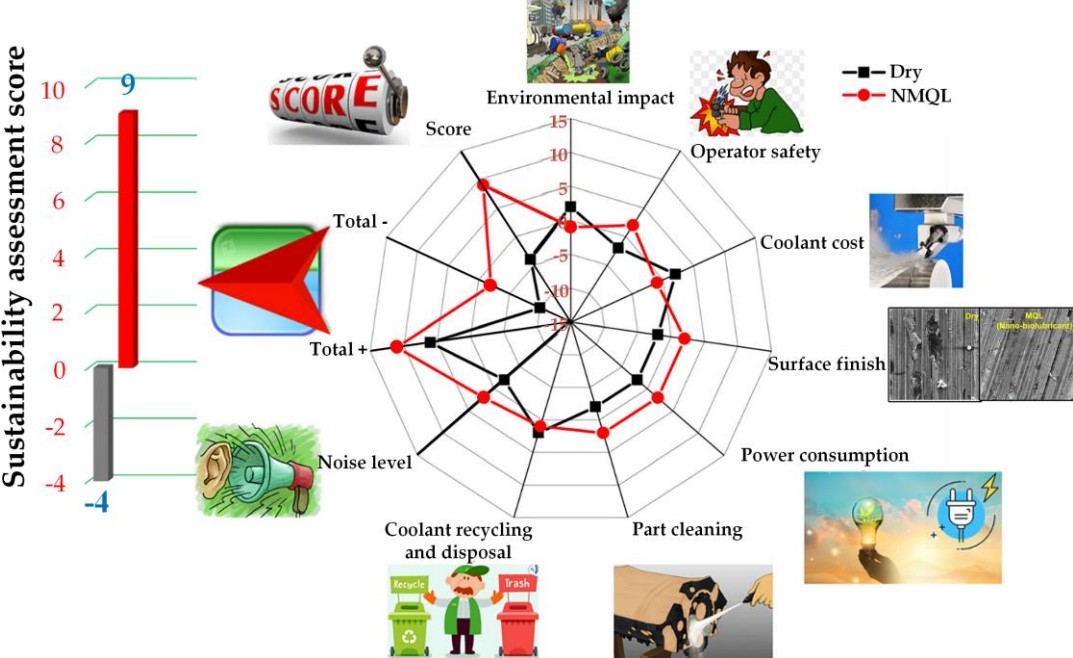

**Figure 26.** Sustainability assessment results for dry and NMQL (reproduced from [269], with permission from Springer Nature).

The preparation of nanoparticles may incur additional energy consumption. However, research shows that using natural materials such as silica, alumina and iron oxide has a much lower environmental impact, because no synthesis is required to produce these particles. Natural nanoparticles are usually non-toxic types, which further reduces the potential toxicity of nanofluids during application and when discharged into the environment [266]. In addition, vegetable oil-based and water-based cutting fluids have achieved good results in terms of environmental and health-related issues by replacing mineral oil-based cutting fluids, achieving high-performance machining and good environmental compatibility [270]. However, the use of new cutting fluids will also be accompanied by new problems, such as the thermal stability and oxidation stability of vegetable oil-based cutting fluids in high-temperature cutting environments. Future research should focus on overcoming the new problems caused by new cutting fluids.

## 6. Conclusions and Prospects

### 6.1. Conclusions

This paper reveals the modeling anti-wear mechanism of NMQL and its application progress in the field of cutting manufacturing. This paper establishes a mapping relationship between material processing and nanofluid selection. This paper provides theoretical guidance and technical support for clean and precision manufacturing technology under the trend of cutting fluid reduction.

- Nano-cutting fluid has excellent cooling and lubrication properties and can be used as an efficient cooling and lubricating medium under the development trend of cutting fluid reduction. Vegetable oil-based and water-based cutting fluids are ideal base fluids for nano-cutting fluids due to their high performance and environmental protection. Hybrid nano-cutting fluids can achieve further enhancement of nanofluid processing performance due to their physical synergistic effects. The dispersion stability of nano-cutting fluid is a key physical property that affects its physical properties and processing performance. The electrostatic stability and steric stability of nano-cutting fluid can be achieved by regulating the dynamic behavior of nanoparticles in the basic fluid through mechanical/chemical dispersion.

- The addition of nanoparticles increases the dynamic viscosity of the fluid, thereby improving the lubricating performance of the cutting fluid. Nanofluid has higher thermal conductivity and specific heat capacity. The addition of nanoparticles reduces the surface tension of the fluid, making it easier for the atomized droplets to penetrate into the capillary in the cutting area; the decrease in surface tension leads to a smaller contact angle, a larger spreading area of the droplet and easier rupture. Compared with dry cutting, the cutting force is reduced by 15–40% under NMQL conditions. The cutting temperature is reduced by 13–76%, and the surface roughness is increased by 10–45%.

- Compared with traditional lubricants, nanoparticles entirely transform the friction mechanism of the lubricant at the -tool–workpiece interface through rolling, filling, film-forming, and polishing effects. Simultaneously, through Brownian motion and heat channels, the heat transfer efficiency of the lubricant experiences a significant increase. The contributions of nanoparticles to reducing the cutting force, decreasing surface roughness, minimizing tool wear, and lowering cutting temperature are 8.6%, 19.2%, 26.8%, and 13.77%, respectively. The contribution of the base fluid is 72.86%, 22.5%, 46.6%, and 67%, respectively.

- Due to the differences in the structure and mode of action of nanoparticles, different particles have different emphasis on performance enhancement. For materials with poor thermal conductivity and sensitivity to heat accumulation, it is more appropriate to prepare nanofluids with excellent cooling properties, such as GR and MWCNTs. When dealing with high-hardness materials, ND nanofluids exhibit excellent polishing ability. After balancing the principles of performance enhancement and economic feasibility, affordable and durable nanoparticles such as $Al_2O_3$ and $TiO_2$ have been

widely used. The infiltration and film formation process of droplets is affected by the surface tension and viscosity of nano-cutting fluid. In continuous cutting, nano-cutting fluids with a faster droplet migration speed should be preferred; in intermittent cutting, the droplet diffusion area and oil film should be optimized.

### 6.2. Prospects

NMQL technology is an environmentally friendly alternative to traditional cutting fluid pouring processing. Through the revelation of its physical mechanism, it can provide broader technical support for the manufacturing of core components in clean precision manufacturing, including aerospace, rail transportation, offshore equipment, engineering machinery, agricultural machinery equipment, power machinery and other fields. Future development trends may focus on the following points:

- Due to structural variations in nanoparticles, nanofluids often exhibit enhanced performance in specific aspects. Blending nanofluids can compensate for these limitations, and through synergistic effects among particles, can further improve performance improvement effects. Seeking multifunctional nanofluid formulations is a key direction. The high-pressure assist provided by MQL systems overcomes the air barrier caused by the high-speed rotation of cutting tools. However, traditional pneumatic atomization is not stable; the dispersion of droplets into the air continues to pose a significant health risk. Electrostatic atomization, which employs electric field forces to constrain droplet dispersion, is a pivotal research direction for future atomization methods.

- Extensive and in-depth research should be carried out into multi-field empowering equipment including ultrasonic empowering, low-temperature empowering, magnetic field empowering and textured tool-assisted nano-fluid MQL cutting and grinding-, and the relationship between process parameters and optimal enabling parameters should be determined. Mapping relations: First, solve the problems of sensor connection of production equipment/production lines, collection, storage and construction of production data, and analysis and mining of production-related big data. Intelligent follow-up nozzles need to be applied on a large scale to achieve the optimal spray posture of the nozzle and online adjustment of process parameters. In addition, as a means of improving lubrication parameter control strategies, communication technology between machine tools and MQL supply systems also requires large-scale industrial applications.

- Due to machine tool operators' insufficient understanding of the action rules of nano-cutting fluid, it is difficult to achieve precise control of processing parameters during the industrialization process. Therefore, there is an urgent need to establish a standardized public database, including workpiece materials, lubricating medium parameters (such as the chemical formula of the base fluid, nanoparticles/size/shape/concentration, supply flow rate, atomization pressure, nozzle distance and angle), cutting process parameters (cutting speed, feed and depth of cut) and machining performance evaluation (such as cutting forces, the cutting temperature, tool life and workpiece surface integrity). In addition, the mapping relationship between cutting parameters, lubricating medium parameters and evaluation indicators should be established. On this basis, an artificial intelligence software platform for recommending NMQL process parameters can be further built based on neural network algorithms.

**Author Contributions:** Investigation, writing—original draft, and writing (review and editing), A.C.; technical and material support, instructional support, and writing—review, C.L.; collect and organize data and writing—review and editing, Z.Z.; formal analysis and validation, B.L.; formal analysis and validation, Y.Z.; modify paper and formal analysis, M.Y.; collect and organize data, T.G.; modify paper and validation, M.L.; modify paper and validation, N.Z.; formal analysis and validation, Y.S.D.; conceptualization, formal analysis, S.S. All authors have read and agreed to the published version of the manuscript.

**Funding:** This research was funded by the National Key Research and Development Program, China (2020YFB2010500), National Natural Science Foundation of China (52375447, 52105457, 51975305), the Special Fund of Taishan Scholars Project (tsqn202211179), the Youth Talent Promotion Project in Shandong (SDAST2021qt12), the Natural Science Foundation of Shandong Province (ZR2023QE057, ZR2022QE028, ZR2021QE116, and ZR2020KE027) and the Qingdao Science and Technology Planning Park Cultivation Plan (23-1-5-yqpy-17-qy).

**Conflicts of Interest:** The authors declare no conflict of interest.

## Abbreviations

| | | | |
|---|---|---|---|
| MQL | minimal quantity lubrication | C=C | carbon-carbon double bond |
| NMQL | nanofluid minimum quantity lubrication | $C_p$ | specific heat capacity |
| CFRP | carbon fiber-reinforced plastics | K | thermal conductivity |
| ND | nano-diamond | μ | viscosity |
| SWCNT | single-walled carbon nanotube | CCF | commercial cutting fluids |
| DWCNT | double-walled carbon nanotube | SME | spatially multiplexed exposure |
| MWCNT | multi-walled carbon nanotube | MMC | metal matrix composites |
| GR | graphene | EG | ethylene glycol |
| CNT | carbon nanotube | RC | relative concentration |
| DW | deionized water | | |

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
