# Peer review of "Nanofluids Minimal Quantity Lubrication Machining: From Mechanisms to Application"

_lubricants, doi:10.3390/lubricants11100422_

Round 1
Reviewer 1 Report
In this paper, the authors reviewed and analyzed prior research on Nanofluids minimal quantity lubrication machining: from mechanisms to application. The most relevant results of the evaluated studies were presented to aid researchers in understanding the advances in Nanofluids minimal quantity lubrication machining: from mechanisms to application.
The research is of great interest. This manuscript has some weaknesses and here are the most important topics/questions to be dealt with:
1.Please rewrite the abstract by identifying the purpose, the problem, the methodology and the important results (not all) and conclusions of your work.
2. In Introduction section:
The Introduction should consist of five paragraphs answering the following five questions:
What is the problem?
Why is it interesting and important?
Why is it hard?
Why hasn't it been solved before? (Or, what's wrong with previously proposed solutions?)
What are the key components of my approach and results?
3.The originality of the paper needs to be stated clearly. It is of importance to have sufficient results to justify the novelty of a high-quality journal paper.
4.You must add a table of nomenclature and abbreviations.
5.The Conclusion should summarize the work of the whole paper, rather than simply summarize the results. Furthermore, the advantages and drawbacks of the proposed method should be highlighted. The future works about this subject should be pointed out in the part.
Happy to give more support shortly.
Reviewer 2 Report
In recent years, environmentally friendly production activities have become the main target for the manufacturing industry. The oils used during the machining of hard-to-machine parts in the manufacturing industry harm the environment, and these oils should be minimized. This study aims to provide a roadmap for the preparation of nanofluids, the determination of their physical and chemical properties, and especially for the determination of optimum nano-cutting fluid parameters for the machining of different materials.
Editing English lines 24,51,60 etc.
"Van der Waal" should be corrected to Van der Waals.
Third-person singular pronouns should be preferred in scientific articles. It is recommended that the article be revised in this context.
To improve the quality of the study, it is recommended that the following items be answered and done.
-Why are the results of the shear test of coconut oil-based molybdenum dioxide NMQL included in the abstract, and why is only this experiment mentioned?
The introduction is well organized, but more studies would increase the study's target audience.
-Zhou et al. [46] in line 127 is not mentioned in the references section. [147] will be organized as Minakov. [208] should be corrected to Huang et al. It is recommended to check the references and citations one by one.
The image quality of the figures should be improved. Some text is not readable.
Editing English lines 24,51,60 etc. Since it is a review study, it is recommended to double-check the whole article.
"Van der Waal" should be corrected to Van der Waals.
Third-person singular pronouns should be preferred in scientific articles. It is recommended that the article be revised in this context.
Reviewer 3 Report
I believe that the article is written with care and provided the results that are essential for a nice manuscript. I have some minor suggestions
1. The authors should compare their work with the recent work in literature, and provide the gap of the research.
2. How this work is beneficial in the field, provide solid remarks regarding this.
3. The graphical results are fine however some more explaination are requested.
4. The mathematical equations should be explained in details
Reviewer 4 Report
The presented review paper is interesting to read. There is a lot of collected and summarised information. I believe it will be helpful to the reader.
By reading, I expected to find more profound information on the preparation of nanofluids. However, there is only general information.
The conclusion and prospects could be more oriented to the trends of nanomaterials. In the present case, the processing techniques are more emphasised. Moreover, the statement in Line 1280 „The use of textured cutting tools can significantly enhance the cutting performance of challenging-to-machine materials.“ is out of the topic.
Why is the experiment mentioned in Lines 28 to 30 essential to present in the abstract?
There are a few shortages I would suggest to clarify:
- Is the terminology correct in Figure 2? „Nanofluid cutting fluid...“.
- Figure 3 is not mentioned in the text.
- The interpretation of advantages and disadvantages in Figure 3 is slightly confusing. The authors state that environmental and health issues are disadvantages of water-based emulsions. I believe authors have in mind a fully formulated emulsions. On the other hand, vegetable oils-based cutting fluids have the advantage of „Toxic free and safe“. Is it also fully formulated? Moreover, do the fully formulated water-based emulsions have poor lubricity? The authors should consider whether they discuss the properties of base fluid or fully formulated one.
- If we consider Figure 3 as a reference, the mineral base oils must also be discussed. Now, only water and vegetable oils are discussed in sections 2.1.1 and 2.1.2.
- There are many correction errors, for instance, in Table 5, „An-ionic surfactant“, „cationic surfactant“.
- I would suggest discussing which stabilisation methods are common in metal working fluid applications.
- There is a statement in Table 7: „The viscosity index at 70°C was only approximately half of that at 40°C, highlighting a substantial downward trend.“ The viscosity index is a constant for every fluid and does not change with the temperature.
- The action mechanisms are discussed on Pages 23-24, Lines 774 to 790. Please clarify the metal working procedure where these mechanisms could appear. For instance, is the third one applicable for turning or milling?
Reviewer 5 Report
The paper provides a review of nanofluids and their application in minimum quantity lubrication (MQL) machining, covering their preparation, properties, mechanisms, and performance across different materials. The review covers the important aspects of nanofluid preparation, properties and application in MQL comprehensively with 267 references. The paper is well-organized and clearly written. I recommend minor revisions addressing the below aspects and highlighting practical insights before publication.
· More details could be provided on specific nanoparticle concentrations and base fluid mixing ratios found optimal in different studies. How were these optimized?
· The section on stability of nanofluids is quite extensive. Are there any common techniques used to improve stability in the studies reviewed?
· The authors provide no discussion on standardization of nanofluid preparation or characterization across the literature. Results may vary due to differences in synthesis methods.
· Lacks a detailed economic analysis or life cycle assessment of nanofluids. Viability for industry adoption is not explored in depth.
· Role of base fluids and interaction effects with nanoparticles are not fully explored. Focus is more on nano-effects.
· Including more quantitative data in tables would help summarize the reductions achieved with nano-MQL across different materials and studies.
· The performance improvements are attributed to nanoparticles, but interaction effects between base fluid properties and nanoparticles could be analyzed more.
· Broader challenges with nanofluid machining like cost, disposal and health/safety considerations could be discussed.
· Recent advances like magnetically responsive nanofluids and future research directions could provide a more forward-looking perspective.
English Language Issues
There are some grammatical mistakes in the paper. Some of the key grammar issues are incorrect subject-verb agreement, missing words, incorrect pluralization, capitalization errors, and lack of articles in some sentences. Overall, the paper would benefit from further proofreading and grammar correction. Some of the mistakes are listed below:
Page 2, Line 52: Dragičević 's survey find … should be “Dragičević 's survey fond”
Page 2, Line 68: space is missing in “effective cooling means,it”
Page 6, Line 180: C=C double bond has the capability to induce curvature in a carbon chain. .. should start with “The C=C”
Page 19, Line 635: Brownian motion and nanofluid cluster aggregation effect .. …. Should be corrected as “effects”
Minor improvement required
